# Alveolar macrophage-expressed Plet1 is a driver of lung epithelial repair after viral pneumonia

Learta Pervizaj-Oruqaj [1,2,3,12], Balachandar Selvakumar [1,4,5,12], Maximiliano Ruben Ferrero [1,2,3,4,5,12], Monika Heiner[1,2,3], Christina Malainou [1,2,3], Rolf David Glaser [2,6], Jochen Wilhelm [2,3,7], Marek Bartkuhn[2,6], Astrid Weiss [3,7], Ioannis Alexopoulos [1,2,3], Biruta Witte[8], Stefan Gattenlöhner[9], István Vadász [2,3,7], Rory Edward Morty [10], Werner Seeger [2,3,4,5,7], Ralph Theo Schermuly [2,3,7], Ana Ivonne Vazquez-Armendariz [1,2,3,11] & Susanne Herold [1,2,3] ✉

Influenza A virus (IAV) infection mobilizes bone marrow-derived macrophages (BMDM) that gradually undergo transition to tissue-resident alveolar macrophages (TR-AM) in the inflamed lung. Combining high-dimensional single-cell transcriptomics with complex lung organoid modeling, in vivo adoptive cell transfer, and BMDM-specific gene targeting, we found that transitioning ("regenerative") BMDM and TR-AM highly express *Placenta-expressed transcript 1* (Plet1). We reveal that Plet1 is released from alveolar macrophages, and acts as important mediator of macrophage-epithelial cross-talk during lung repair by inducing proliferation of alveolar epithelial cells and re-sealing of the epithelial barrier. Intratracheal administration of recombinant Plet1 early in the disease course attenuated viral lung injury and rescued mice from otherwise fatal disease, highlighting its therapeutic potential.

Tissue-resident alveolar macrophages (TR-AM) are long-lived cells[1] localized to the alveolar space, where they exert key functions in maintaining tissue homeostasis and in immediate host defense. Several studies revealed that TR-AM are depleted upon viral pneumonia and gradually replaced by BMDM (CD11c⁺CD11b⁺Ly6C⁺CX₃CR1⁺) during the infection course, eventually resulting in reprogramming of the TR-AM pool[1–4]. BMDM are recruited to sites of infection via CCR2 ligation[5]

and are considered to reveal high functional plasticity. Polarization of BMDM into different macrophage phenotypes in the infected lung is thought to be driven by integration of spatially and timely resolved signals from the inflamed microenvironment or tissue niche in a disease-specific manner[6,7]. Whereas pro-inflammatory macrophages were found to contribute to the severity of viral pneumonia by directly damaging the lung parenchyma, and to aberrant lung remodeling[8–11],

[1]Department of Internal Medicine V, Universities of Giessen and Marburg Lung Center, University Hospital Giessen, Justus Liebig University, Member of the German Center for Lung Research (DZL), Giessen, Germany. [2]Institute for Lung Health (ILH), Justus Liebig University, Giessen, Germany. [3]Excellence Cluster Cardio-Pulmonary Institute (CPI), Giessen, Germany. [4]Max Planck Institute for Heart and Lung Research, Bad Nauheim, Germany. [5]Instituto de Investigación en Biomedicina de Buenos Aires (IBioBA), Buenos Aires, Argentina. [6]Biomedical Informatics and Systems Medicine, Justus Liebig University, Giessen, Germany. [7]Department of Internal Medicine II, Universities of Giessen and Marburg Lung Center, University Hospital Giessen, Justus Liebig University, Member of the German Center for Lung Research (DZL), Giessen, Germany. [8]Department of General and Thoracic Surgery, University Hospital of Giessen, Giessen, Germany. [9]Department of Pathology, University Hospital of Giessen, Giessen, Germany. [10]Department of Translational Pulmonology and the Translational Lung Research Center, University Hospital Heidelberg, Member of the German Center for Lung Research (DZL), Heidelberg, Germany. [11]University of Bonn, Transdisciplinary Research Area Life and Health, Organoid Biology, Life & Medical Sciences Institute, Bonn, Germany. [12]These authors contributed equally: Learta Pervizaj-Oruqaj, Balachandar Selvakumar, Maximiliano Ruben Ferrero. ✉e-mail: Susanne.herold@innere.med.uni-giessen.de

the role of distinct macrophage subsets in lung repair, and in particular their effector molecules mediating cross-talk with lung epithelial (stem/progenitor) cells during such processes are poorly understood. Nevertheless, mounting evidence supports that recruitment and expansion of macrophages are necessary to promote tissue regeneration after injury[12]. Efforts to elucidate the molecular mechanisms by which macrophages support the expansion of epithelial (stem/progenitor) cells, increase their resilience towards injury and infection, or halt their apoptotic death aim to prevent the rapid decline of gas exchange function associated with severe pneumonia, and to accelerate its re-establishment through coordinated regeneration processes. Indeed, a recent study revealed airway epithelial repair driven by an IL-33/ST2-induced macrophage differentiation program, and signals activating the IL-4 receptor on BMDM accelerated de novo lung tissue formation after pneumonectomy[13,14].

Placenta-expressed transcript 1 (Plet1) is a 207 amino acid glycophosphatidylinositol (GPI)-anchored protein with unknown receptor, expressed in proliferating epithelia and in stem cell niches, such as differentiating trophoblasts[15,16], follicular and thymic stem cells[17–19]. It is involved in keratinocyte migration[20] and gut epithelial wound healing from the Lgr5[+] colon stem cell niche[21]. We and others recently identified its expression in myeloid cells and its role in dendritic cell migration and in the instruction of innate lymphoid cells to release IL-22[22–24] (data accessible at NCBI GEO database, GEO accession GDS1874). However, its expression pattern and function in the context of lung injury and regeneration have been unknown.

Here, we reveal that during the resolution phase of viral pneumonia a transitional subpopulation of BMDM and the re-emerging TR-AM pool constitute key components of epithelial stem/progenitor cell niches and contribute to lung regeneration after injury through the expression of Plet1. Using bone marrow chimeric mice and single-cell transcriptome data, we confirmed the spatiotemporal kinetics of the BMDM-to-TR-AM transition, and captured that this trajectory is associated with distinct functional phenotypes from inflammatory towards tissue-regenerative, characterized by gradual increase of Plet1 expression. Adoptive transfer of Plet1[+] macrophage populations sampled along this differentiation trajectory, into Ccr2[-/-] mice, combined with conditional CX₃CR1[iCRE]-Plet1[flox/tom] transgenic mouse and lung organoid assays modeling the macrophage-epithelial niche during alveologenesis, revealed that Plet1[+] macrophages efficiently attenuate influenza A virus (IAV)-induced epithelial injury and foster stem/progenitor cell-driven alveolarization and barrier repair. Finally, intraalveolar administration of recombinant Plet1 rescued mice from fatal viral pneumonia, highlighting its potential as a therapeutic to combat severe inflammatory lung injury in viral pneumonia.

## Results

### Alveolar macrophages cluster into TR-AM and CD40[high] inflammatory versus CD206[high] transitional BMDM subsets that reveal a regenerative phenotype in IAV-infected mice

Macrophage-epithelial interactions during epithelial repair were suggested to widely occur in the bronchoalveolar airspace compartment[25,26]. To map the dynamics of macrophage composition and phenotypes in the intraalveolar compartment of the murine lung after IAV infection, flow-sorted CD11c[+] bronchoalveolar lavage fluid (BALF) cells were profiled by single-cell (sc) RNA-seq at the indicated time post infection (p.i.), capturing the alveolar injury and resolution phase of the infection (Fig. 1a and Supplementary Fig. 1a). Unsupervised clustering over all time points distributed cells into 8 transcriptionally distinct clusters of CD11c[+] cells (Fig. 1b). As shown in Fig. 1c (depicting the 3–5 top differentially expressed genes in each cluster) and Fig. 1d (depicting selected/canonical marker genes), we identified a population of antigen-presenting dendritic cells (DC; expressing H-2 MHC II genes, Ccr7, and the conventional DC1 marker Zfp366, coding for DC-SCRIPT[27]) and a cluster of CD11c[+] cytotoxic T

cells (expressing Cd3d, Cd3e, Cd8a, Ccr5, and granzymes (Grzma/Grzmb)[28]. TR-AM were characterized by expression of Siglecf, Ear2, Mertk, and Fabp1 (see also Supplementary Fig. 1b), and revealed two low-frequent proliferating subclusters (cycling TR-AM 1 and 2, characterized by genes associated with DNA replication, Mcm3-6, and by the proliferation marker Mki67, respectively). Another low-frequent TR-AM population expressed antiviral genes dependent on type I interferon signaling, such as Tnfsf10, Ifit1b1, and Cmpk2[29,30] termed Cmpk2[hi] TR-AM. Of note, two different clusters of BMDMs were identified (BMDM1/2). Compared to BMDM2, BMDM1 were Ly6c2[high] and expressed higher levels of transcripts related to immediate immune response and inflammation, such as Apoe, complement-associated genes C1qa and C3ar1, Fcgr1 (coding for the high-affinity Fc gamma receptor), Itgam (coding for CD11b), and of chemokine receptors (Ccr2, Cx3cr1, Ccr5). Resolution over time revealed distinct kinetics of alveolar CD11c[+] cell clusters (for kinetics of TR-AM and BMDM marker genes, see Supplementary Fig. 1b). At D7 p.i., TR-AM (except Cmpk2[hi] TR-AM) were widely depleted and gradually returned between D14 to D35 p.i. BMDM1 appeared in the alveoli at high frequency by D7 p.i., coinciding with the peak alveolar injury in this model[11], and their numbers declined until D35. Conversely, whereas few BMDM2 cells were already present in the airspace at day 7, their numbers increased at D14 and peaked at D21 p.i., coinciding with the phase of injury resolution, alveolar repair and return to homeostasis (Fig. 1e, f)[8]. In line, transcripts associated with antiviral host defense (Irf1, Irf7, Isg15, Ifitm3, Cxcl10), inflammation (MyD88, Irg1, Aif1, Hif1a), chemotaxis (Ccl2, Ccl4, Ccl5) or with epithelial injury induction (Tnfsf10, Tnfa, Il1b) were highly expressed in D7 BMDM1, whereas transcripts associated with alternative activation (Pparg, Klf4, Myc[31], Cav2[32], Rxra[33], tissue remodeling (Fn1, Vim, Tgfb1), and cell growth (Crip1, Pdgfc) were higher in D21 BMDM2 (Fig. 1g), suggesting that BMDM2 might exert distinct functions in tissue regeneration. We next performed gene set enrichment analyses (GSEA) based on gene ontology (GO) terms in BMDM1/2 at D14 and 21 p.i. Differentially enriched GO terms are presented in Fig. 1h, revealing a predominant activation of genes related to inflammation and host defense in BMDM1 at both time points, while genes related to fatty acid metabolism, wnt signaling, tight junction (TJ) organization, and epithelial cell development were enriched in BMDM2. In addition, the BMDM2 cluster expressed genes or transcription factors associated with TR-AM identity (Ear2, Mertk, Spp1, Sort1, Pparg, Klf4; Supplementary Fig. 1b); thus, we hypothesized that this cluster represents a transitional state along the differentiation program into TR-AM driving return to tissue homeostasis[3,34]. This hypothesis was further supported by trajectory inference by RNA velocity analysis of the scRNA-Seq dataset (Fig. 1) and partition-based graph abstraction (PAGA), predicting that BMDM1 transitioned to BMDM2, which in turn, represented an intermediate state towards differentiation into TR-AM clusters (Supplementary Fig. 1 c, d). In line with these data, using CD45.1/2 chimeric mice undergoing total body irradiation, BM transplant, and IAV infection after BM reconstitution (Supplementary Fig. 1e), depleted TR-AM of recipient (CD45.1) phenotype were gradually replenished between D14 and D21 by CD45.2[+] BM-derived precursors, confirming that BMDM are able to replenish the TR-AM pool in this model (Supplementary Fig. 1f, g), as previously reported[3,4,35].

In an effort to correlate transcriptional heterogeneity of BMDM with unique functional phenotypes in the course of IAV infection, we aimed to establish a robust protocol to flow-sort BMDM subsets for further analyses, using distinct surface markers for identification. Out of several surface antigens differentially expressed according to the single cell transcriptome dataset in Fig. 1, we identified CD40 (Cd40) and CD206 (Mrc1) to be most suitable to identify BMDM1 and 2, respectively (Fig. 2a, b; gating strategy in Supplementary Fig. 2a). BMDM sub-phenotyping according to CD40[high] and CD206[high] populations by FACS reproduced the BMDM1/2 kinetics identified in the

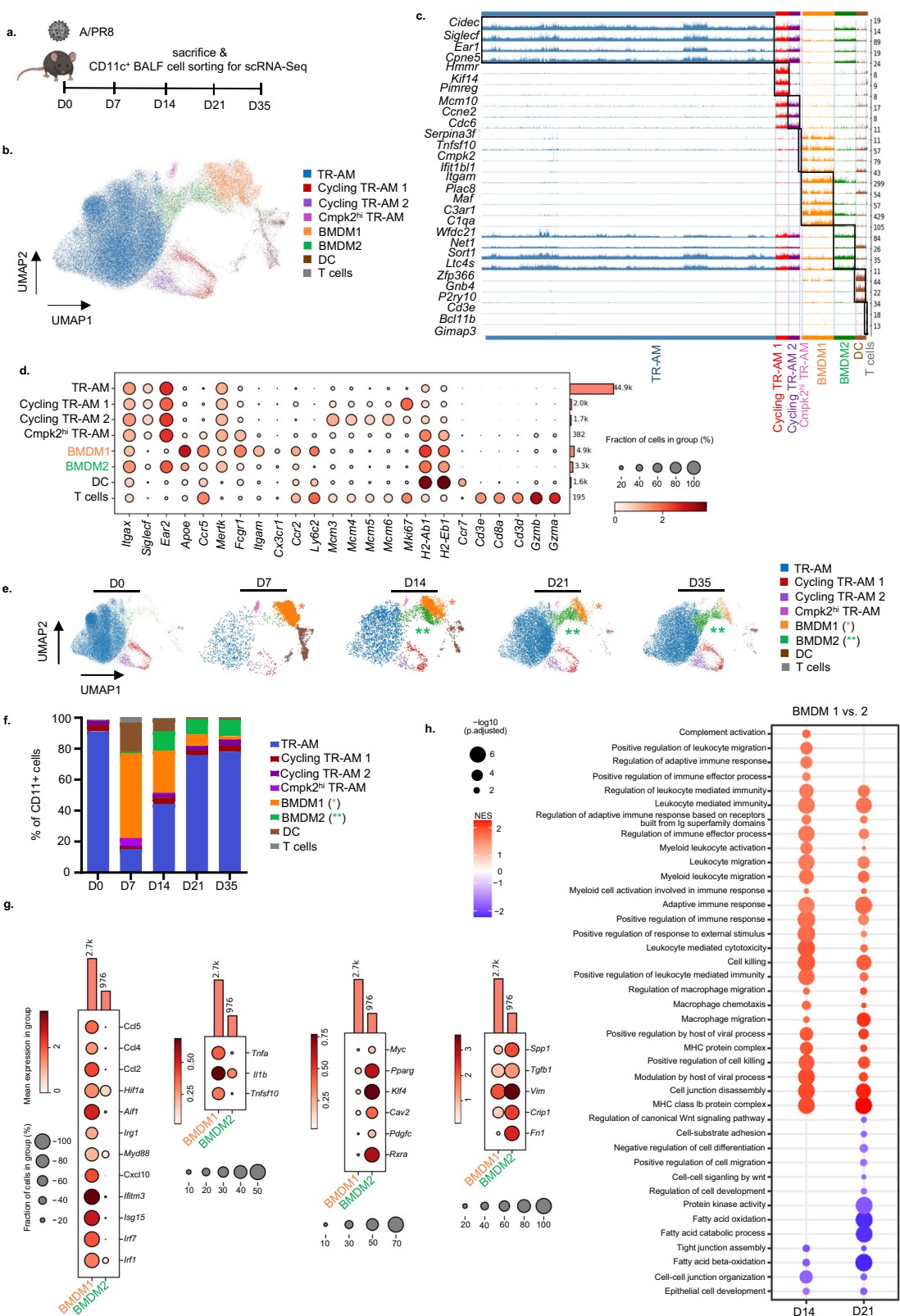

scRNA-seq dataset (Fig. 1e and Supplementary Fig. 2b), with a peak accumulation of CD40[high] BMDM1 at D7, and alveolar presence of CD206[high] BMDM2 by D14 to D21 (Fig. 2c). BMDM kinetics in lung homogenates of pre-lavaged IAV-infected mice revealed similar kinetics as observed in the airspaces for BMDM1. However, BMDM2 appeared earlier in lung tissue, at day 10, and lower numbers of these cells were observed at later time points compared to BALF analyses,

likely capturing their rapid interstitial-to-alveolar transmigration (Supplementary Fig. 2c). To unambiguously verify that this gating strategy corresponded to the respective UMAP clusters of BMDM, we performed transcriptional profiling by cDNA microarray on BALF of flow-sorted CD40[high] versus CD206[high] BMDM, obtained on days 7 and 21 p.i., respectively. Data in Fig. 2d confirmed differential gene expression in CD206[high] BMDM2 (genes associated with alternative

**Fig. 1 | scRNA-seq analyses reveal two distinct alveolar BMDM clusters with defined kinetics of appearance over the course of IAV infection. a** Schematic representation of the infection model and sample collection in uninfected mice (D0) and at the indicated time points p.i. The figure was created with BioRender. **b** Uniform manifold approximation and projection (UMAP) map displaying CD11c⁺ alveolar cell clusters (pooled data from all time points), integrated and embedded using the Harmony algorithm. Colors represent different cell populations after Leiden clustering in 1.4 resolution (58,998 individual cells with signals from 32,286 genes; average number per time point from $n = 3–5$ mice). **c** Track plot showing unbiased analysis of the top three to five expressed genes in each particular cluster; pooled data from all time points. **d** Dot plot representing signature markers for each cell cluster. The size of the dots encodes the percentage of cells within a cluster expressing a particular marker. Color of the dots represents the average expression level throughout all cells within a cluster. **e** UMAP depiction of alveolar

CD11c⁺ cell clusters at the respective time points p.i. **f** Stacked bar chart showing the percentage of CD11c⁺ cells during the infection course. **g** Dot plots displaying expression of selected phenotype markers in BMDM1 and BMDM2 populations. Genes shown in the plot were selected among the significantly higher expressed genes ($p < 0.05$), The Wilcoxon rank sum test was employed as the statistical method. The size of the dots encodes the percentage of cells within a cluster expressing a particular marker. Color of the dots represents the average expression level throughout all cells within a cluster. **h** Gene ontology (GO)-based GSEA showing normalized enrichment scores (NES) for expression differences of genes comparing BMDM1 compared to BMDM2 on D14 and 21 p.i. An array of statistical tests provided by the fgsea package was employed. Data in (**b**)–(**h**) refer to the experiment depicted in (**a**). Tissue-resident alveolar macrophages (TR-AM), bone marrow-derived macrophages (BMDM), dendritic cell (DC), T lymphocyte (T).

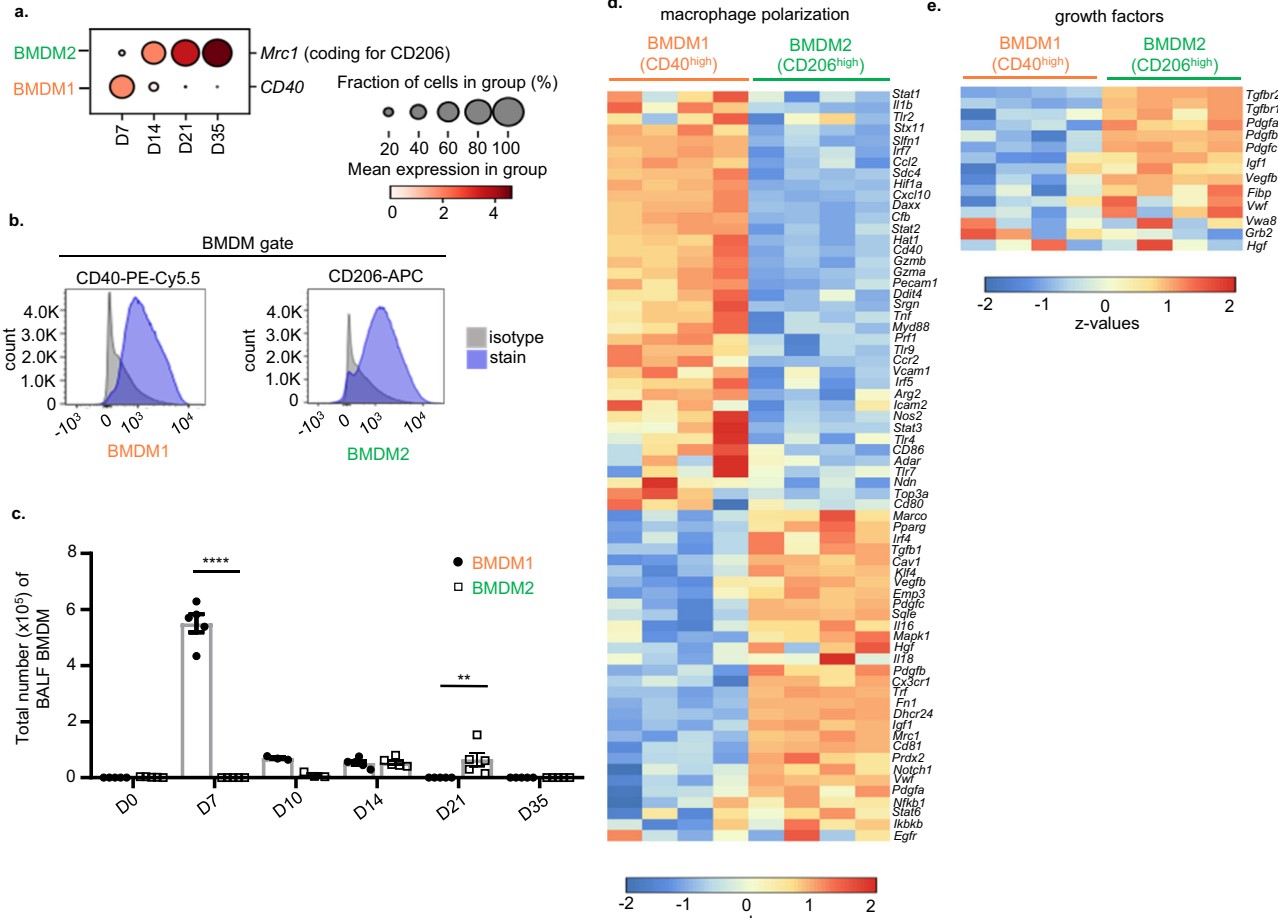

**Fig. 2 | CD40 and CD206 expression identify BMDM1 and BMDM2 clusters in the lungs of IAV-infected mice. a** Dot plot graph showing expression of CD40 and CD206 in BMDM1 and BMDM2, respectively. The size of the dots encodes the percentage of cells within a cluster expressing the particular marker. Color of the dots represents the average expression level throughout all cells within a cluster. Data refer to the scRNA-seq experiment in Fig. 1. **b** Representative FACS histograms displaying CD40 expression in BMDM1 and CD206 expression in BMDM2 collected from BALF of IAV-infected mice on days 7 and 21, respectively. **c** Bar graphs show total BMDM1 and BMDM2 cell counts in BALF samples of mice at the indicated time points. Values are representative of three independent experiments shown as

mean ± SEM with individual data points for each group with $n = 5$, except for D10 samples where $n = 3$. Probability determined using two-way ANOVA (Holm–Sidak's multiple comparisons test) ****$p < 0,0001$, **$p = 0.0043$; for comparison between BMDM1 and BMDM2. **d** Heatmap of DNA microarray profiling depicting genes associated with macrophage polarization phenotypes and **e** with genes associated to epithelial growth factor signaling in CD40^high BMDM1 and CD206^high BMDM2 flow-sorted from BALF of IAV-infected mice at D7 and D21 p.i., respectively; prior gating of BMDM was performed according to Supplementary Fig. 2a. Source data are provided as a Source Data file. BMDM bone marrow-derived macrophage, BALF bronchoalveolar lavage fluid, D day.

macrophage polarization and TR-AM phenotype; e.g. *Pparg, Klf4, Pdgfc, Fn1, Tgfb1*) vs. CD40^high BMDM1 (genes associated with an inflammatory phenotype, e.g., *Cxcl10, Ccl2, Tnfa, Il1b, Hif1a*). The expression of selected genes indicative of d7 BMDM1 and d21 BMDM2 phenotypes by qPCR (*Cd40, IL1b, Inos, Tnfa* in CD40^high d7 BMDM1

versus *Mrc1, Arg1, Fizz1* and *Tgfb* in CD206^high d21 BMDM2; Supplementary Fig. 2d) confirmed the phenotype differences observed in cDNA microarray. When we compared BMDM1/2 expression of the same selected genes on d14, when both BMDM are simultaneously present in the lung, the differences were attenuated, suggesting

transitional phenotypes at that time point (Supplementary Fig. 2d). Of note, we confirmed increased expression of several genes related to epithelial growth factor signaling in CD206[high] BMDM2 (Fig. 2e). Together, these data suggest that BMDM recruited to the airways undergo a sequential progression through two distinct phenotypes, with defined kinetics, following severe IAV infection. In contrast to CD40[high] pro-inflammatory BMDM1, transitional CD206[high] BMDM2 reveal a transcriptional profile associated with tissue regeneration, similar to the TR-AM profile, and are present during the resolution phase of the disease course, contributing to TR-AM pool replenishment.

## BMDM2 as opposed to BMDM1 exert proliferative and barrier-protective effects on alveolar epithelial cells in vitro and in vivo

To investigate whether BMDM transition towards BMDM2 endowed the latter with an epithelial-regenerative phenotype, as suggested by the transcriptome analysis, flow-sorted BALF BMDM subsets were co-cultured with ex vivo IAV-infected murine alveolar epithelial cells (AEC). While BMDM1 (collected at d7 p.i.) increased IAV-induced AEC apoptosis, BMDM2 (collected at d21 p.i.) induced AEC proliferation after infection (Fig. 3a, b). To confirm the latter, we used bronchoalveolar lung organoid (BALO) cultures that reveal proximo-distal patterning with branched airways and alveoli allowing modeling of lung development and epithelial cell differentiation (Supplementary Fig. 3a), and profiling of macrophage-epithelial cell interactions[36] (experimental setup in Fig. 3c). BMDM2 but not BMDM1 supported the generation of BALO (in terms of numbers and size), suggesting that the presence of BMDM2 promoted epithelial stem cell proliferation and organoid differentiation (Fig. 3d–f). To confirm the robustness of the BMDM2 phenotype outside of its local (inflammation-resolving) microenvironment, we developed a short-term infection model and applied in vivo-generated BMDM2 (vs. BMDM1) by intrapulmonary transfer into IAV-infected Ccr2[-/-] mice) lacking BM-monocyte mobilization and BMDMs in inflamed tissues[8], in an effort to prevent severe injury peaking at D7 (Fig. 3g). BMDM2 were far less injury-promoting than BMDM1 (as quantified by AEC apoptosis; Fig. 3h), and induced proliferation of CD45/31[neg]EpCam[low]T1-α[neg] alveolar epithelial progenitor cells (AEC II)[37], resulting in improved barrier function at D7 p.i. (Fig. 3i, j). We demonstrated previously that lack of BMDM recruitment in Ccr2[-/-] mice per se did not substantially impair IAV clearance (compared to WT)[8], and we did not detect differences in IAV titers in BMDM1 versus BMDM2-transferred Ccr2[-/-] mice (Supplementary Fig. 3b), suggesting a minor contribution of BMDM of any polarization phenotype to viral clearance. Notably, the tissue-protective features of BMDM2 were conserved in the "experienced" TR-AM (i.e., partially BMDM-replenished) in this model, as opposed to naive, fetal monocyte-derived TR-AM that were sampled before depletion by the infection (Supplementary Fig. 3c–f). Together, these data confirm the alveolar epithelial-regenerative function of transitional BMDM2 (and of experienced TR-AM) implied by their transcriptomic profile in ex vivo and in vivo models, and confirm functional heterogeneity of macrophage subsets identified by scRNA-seq.

## Tissue-regenerative macrophage clusters are characterized by expression of *Plet1*

To define putative key regulators of epithelial-regenerative macrophage functions, we compared the top 10 DEGs in the transcriptome data of flow-sorted BMDM subsets. We identified *Plet1* (Placenta-expressed transcript 1), a gene that has been previously associated with wound healing in skin and gut[20,21] and encoding a protein with a GPI (glycosylphosphatidylinositol) anchor localized to cell membranes[20], as the top gene expressed in BMDM2 versus BMDM1 (Fig. 4a). In line with the BMDM2 transitional state towards TR-AM differentiation, *Plet1* has been reported to be a marker gene of TR-AM identity[34,38]. Correspondingly, we found *Plet1* highly expressed

in the BMDM2 and TR-AM clusters in the scRNA-seq dataset across all time points (Fig. 4b, c), whereas BMDM1 showed low expression. *Plet1* expression was confirmed by qPCR (Fig. 4d) and corresponding cell surface expression was quantified by FACS (Fig. 4e) in D7 BMDM1, D21 BMDM2 and in D21, experienced, TR-AM. As a GPI-anchored membrane protein, Plet1 can be shed from the cell surface. Quantification of soluble Plet1 in BALF of IAV-infected WT mice revealed a significant increase starting in the injury phase and peaking at D21 throughout the resolution phase until D35, declining at D60 (Fig. 4f). We next quantified Plet1 in BALF of Ccr2[-/-] mice at D7 p.i. after adoptive transfer of naive TR-AM, BMDM1, BMDM2 or experienced TR-AMs (Fig. 3g and experimental setup in Supplementary Fig. 3c). In this model where macrophages are exposed to acutely infected/injured lung tissue (from D3 to D7 p.i.), we observed that BMDM2 and experienced TR-AMs are major sources of soluble Plet1 in BALF (Fig. 4g). These data suggest that Plet1 surface expression on, or release from, regenerative macrophages was induced by an inflamed or injured alveolar microenvironment. To identify whether a soluble mediator released from the injured AEC during infection was involved, Plet1[+] macrophages were stimulated with IAV-infected versus non-infected AEC conditioned medium. In fact, whereas no Plet1 could be detected in supernatants of infected and non-infected AEC in the absence of macrophages, conditioned medium of 12 or 24 h infected AEC but not of non-infected AEC significantly increased Plet1 release by TR-AM compared to baseline, indicating that Plet1 surface expression and/or secretion/ shedding are significantly increased by AEC signals induced by IAV infection (Fig. 4h).

## Plet1 induces AEC proliferation and exerts epithelial barrier-stabilizing effects through signaling via MEK and Src kinases

To corroborate our hypothesis that BMDM2-expressed Plet1 supported AEC (re-)generation, we co-cultured BALO with BMDM2 in the presence of a neutralizing anti-Plet mAb or isotype control (ctl), or without BMDM (Fig. 3c setup). Both organoid numbers and size were significantly increased in the presence of Plet1[high] BMDM2 (ctl), and this effect was abolished with anti-Plet1 mAb at D7 of culture (Fig. 5a–c) and correspondingly, the alveoli number in fully developed BALO was reduced (Fig. 5d, a). Given that the putative receptor of Plet1 and its downstream signaling events are unknown, we used recombinant (r) Plet1-treated, ex vivo cultured murine AEC for phosphoproteome profiling using a peptide-based kinase activity screen that allows for robust analyses of phospho-tyrosine (PTK) and serine/threonine kinase (STK) activity[39] (Supplementary Fig. 5a). Among the most highly regulated kinases, the screen predicted five kinases of the non-receptor tyrosine family Src to have reduced activity, and five kinases to have increased activity, among them STK and CK2a1 with a known role in cell growth/proliferation and suppression of apoptosis[40], the atypical mitogen-activated protein (MAP) kinase ERK7 and three STKs, a-Raf, b-Raf, and RSK3, all involved in cell growth and survival (Fig. 5e). These data suggest that a Raf-MEK-ERK signaling pathway might be activated in response to rPlet1 stimulation in AEC II (Fig. 5f). Indeed, we confirmed phosphorylation of c-Raf and ERK1/2 in response to rPlet1 stimulation in AEC (Supplementary Fig. 5b–d), and an increase in AEC proliferation in absence, but not in presence of the MEK inhibitor U0126 (Fig. 5g).

Given the beneficial effects of Plet1[high] BMDM2 compared to BMDM1 on alveolar barrier function in vivo (Fig. 3i), we next addressed further putative mechanisms of rPlet1-mediated improvement of AEC barrier function in the context of IAV injury and analyzed tight junction gene expression. IAV infection of ex vivo cultured murine AEC reduced the mRNA expression of *Tjp-1* (encoding zonula occludens-1, ZO-1), but not of tight junction genes *Cldn1* or *Ocln* (encoding claudin-1 and occludin). Treatment with rPlet1 increased expression of all three genes compared to non-treated controls (Fig. 5h), and increased

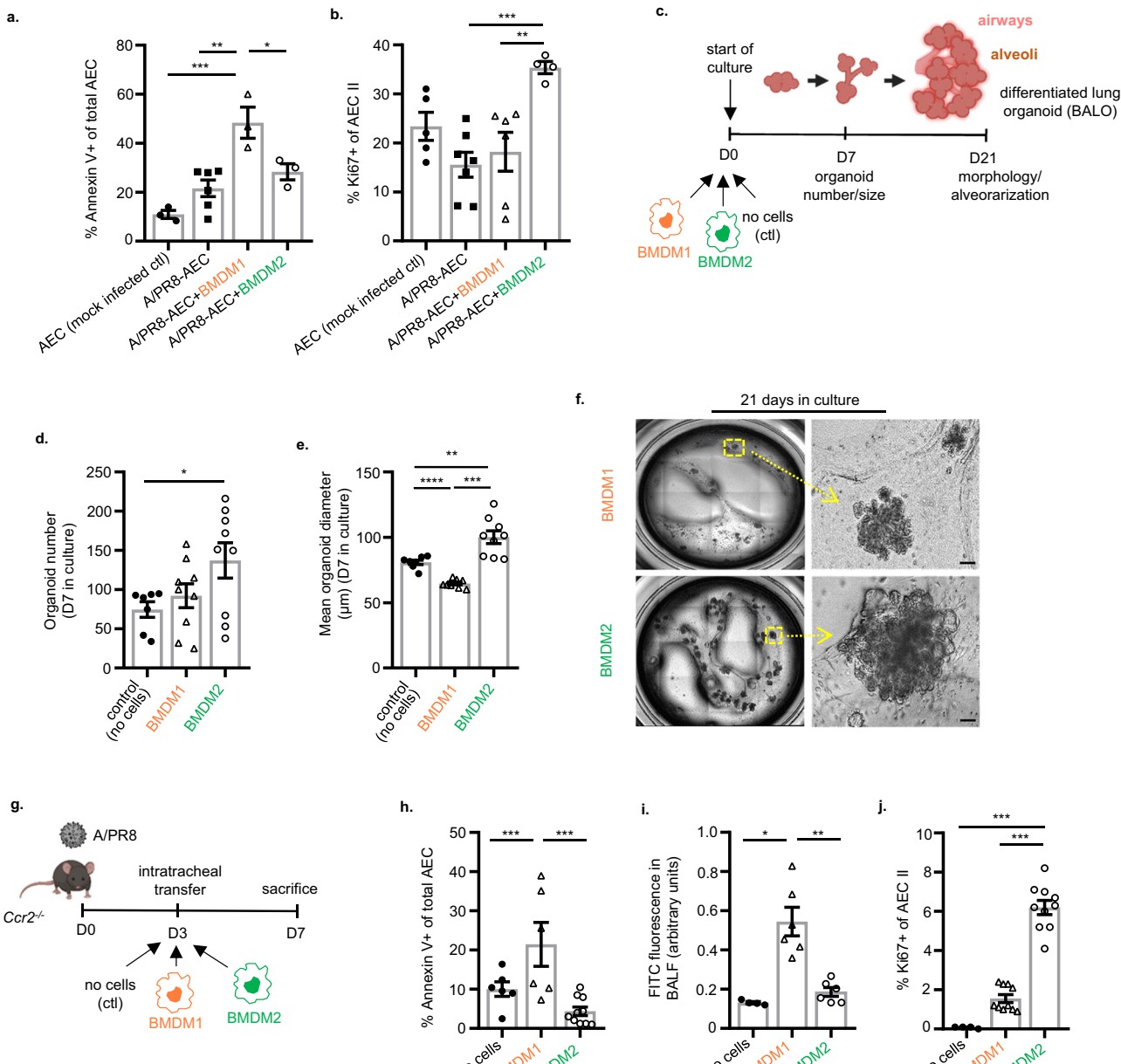

**Fig. 3 | BMDM2 as opposed to BMDM1 exert proliferative and barrier-protective effects on alveolar epithelial cells in vitro and in vivo.** Bar graphs indicate the percentage of apoptotic (**a**, Annexin V+) (*$p = 0.038$, **$p = 0.0023$, ***$p = 0.0005$), and proliferating (**b**, Ki67+) (**$p = 0.008$, ***$p = 0.0018$) control ($n = 3$; 5) or IAV-infected ($n = 6$; 7) (24 h) primary murine AECs that were co-cultured with BMDM1 ($n = 3$; 6) or BMDM2 ($n = 3$; 4) (flow-sorted from BALF of infected mice) for further 24 h. **c** Schematic representation of BALO-BMDM co-culture assay setup. **d** Quantification of organoid numbers (*$p = 0.041$) and **e** diameter (μm) (**$p = 0.0099$, ***$p = 0.0002$, ****$p < 0.0001$) of BALO after 7D of co-culture, in control experiments ($n = 7$) (no BMDM) and in co-culture with BMDM1 ($n = 9$) or BMDM2 ($n = 9$). **f** Representative whole-well picture of organoids after 21D in culture. Images were obtained using EVOS FL microscope. Scale bars indicate 50 μm. **g** Schematic representation of experimental setup of BMDM intrapulmonary transfer into IAV-infected $Ccr2^{-/-}$ mice referring to data in (**h**)–(**j**). The figure was created with BioRender. **h** Percentage of apoptotic AEC (CD31/ 45$^{neg}$EpCam$^+$Annexin V$^+$) analyzed by FACS. Groups: no cells ($n = 6$), BMDM1 ($n = 6$),

BMDM2 ($n = 10$) (*$p = 0.049$, **$p = 0.0012$). **i** Assessment of lung barrier permeability by FITC fluorescence in BALF of $Ccr2^{-/-}$ recipient mice after intravenous application of FITC-labeled albumin, ratios of BALF to serum fluorescence are given as arbitrary units. Groups: no cells ($n = 4$), BMDM1 ($n = 6$), BMDM2 ($n = 6$) (**$p = 0.0079$, ***$p = 0.0053$). **j** Percentage of proliferating AEC II (CD31/ 45$^{neg}$EpCam$^+$T1α$^{neg}$Ki67$^+$) analyzed by flow cytometry. BMDM1 and BMDM2 were flow-sorted from BALF of IAV-infected WT mice at D7 and D21, respectively for experiments in (**a**)–(**j**). Groups: no cells ($n = 4$), BMDM1 ($n = 10$), BMDM2 ($n = 10$) (***$p < 0.0001$, *$p = 0.0236$). Bar graphs are representative of three independent experiments showing means ± SEM and individual data points. Statistical significance was calculated using one-Way ANOVA and Tukey's post-hoc tests except in (**d**) where Dunnett's post-test was used, and (**e**), (**i**) where Brown Forsythe and Welch ANOVA followed by Games−Howell's test was used. In (**d**) and (**e**), single data points represent means of organoid numbers and diameters per well. BMDM bone marrow-derived macrophage, ACE alveolar epithelial cell, BALO bronchoalveolar lung organoid, D day. Source data are provided as a Source Data file.

protein levels of ZO-1 as revealed by immunofluorescence (Fig. 5i). Correspondingly, transepithelial resistance of primary murine and human AEC monolayers was significantly impaired by IAV infection, and this was rescued at least in part by rPlet1 treatment (Fig. 5j). In line

with the findings in Fig. 5e, Src kinase Src, Fyn and Lyn (but not Yes) gene expression were downregulated upon rPlet1 stimulation (Supplementary Fig. 5e), and in turn, Src kinase activation abolished the rPlet1-mediated upregulation of TJ gene expression (Fig. 5k). These

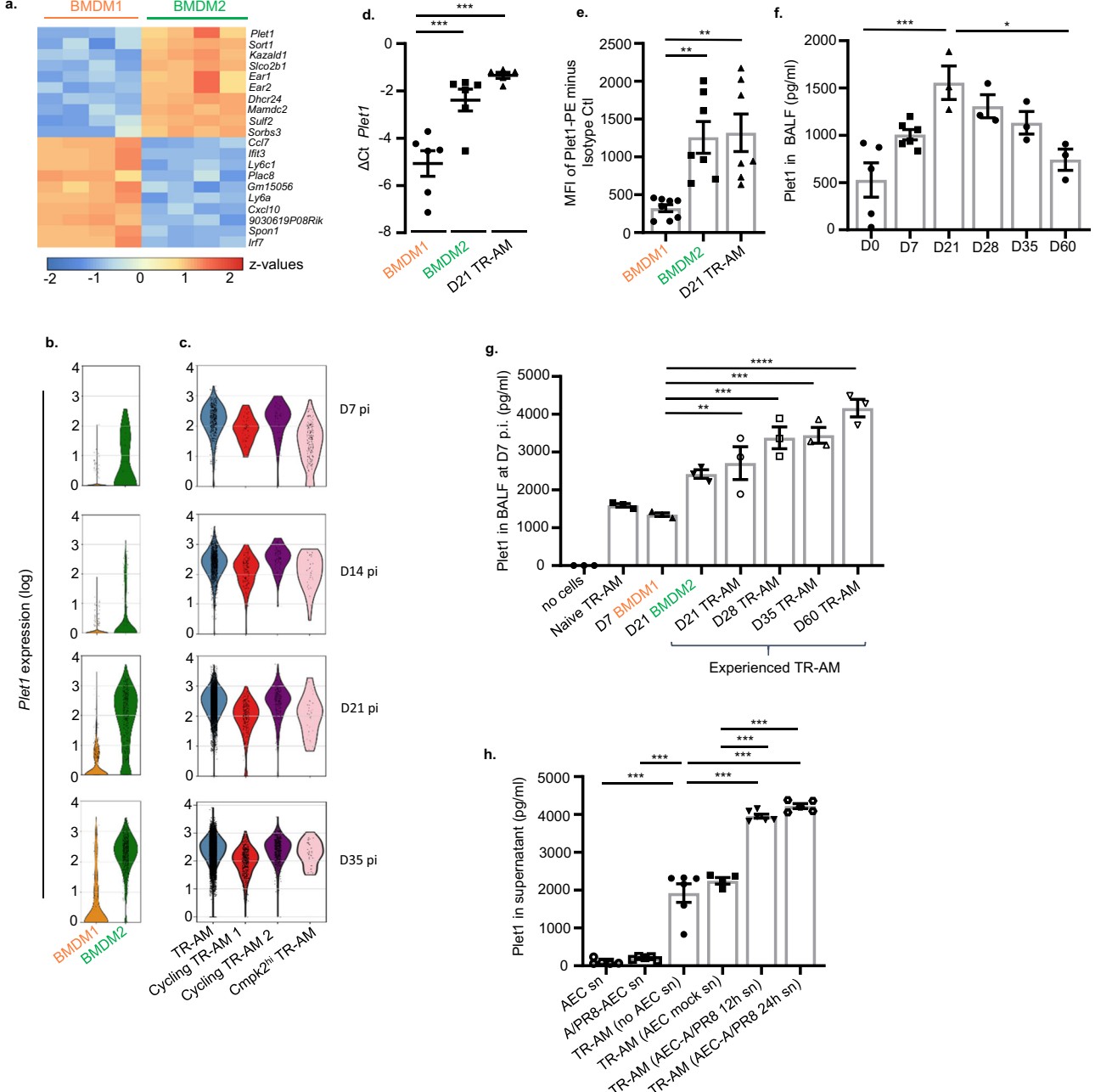

**Fig. 4 | BMDM2 and TR-AM are characterized by increased Plet1 expression and release compared to BMDM1. a** Heatmap of cDNA microarray analysis showing top 10 DEG in BMDM2 versus BMDM1 flow-sorted at D7/D21 p.i., respectively. Genes shown in the heatmap were selected according to *p* value (*p* < 0.05). **b** Violin plots showing expression level (log) of *Plet1* in BMDM and **c** in TR-AM of IAV-infected mice at D7, D14, D21 and D35. Colored areas indicate density distribution in each cluster. Data refer to scRNA-seq experiments in Fig.1b. **d** Comparative *Plet1* mRNA expression analysis by qPCR in BMDM1 of IAV-infected mice obtained at D7; in BMDM2 and TR-AM at D21 p.i. Results are expressed as ΔCt value (Ct reference−Ct target) (*n* = 6) (\*\**p* = 0.0016, \*\*\**p* = 0.0001). **e** Mean fluorescence intensity (MFI) of Plet1 (PE) analyzed by FACS in BMDM1 (D7 p.i.) (*n* = 8), BMDM2 (D21 p.i.) (*n* = 7) and TR-AM (D21 p.i.) (*n* = 7) from BALF of IAV-infected mice (\*\**p* = 0.0042, \*\*\**p* = 0.0025). **f** Quantification of soluble Plet1 in BALF of IAV-infected WT mice at

D0/7/21/28/35/60 p.i. *n* = 3 except on D0 where *n* = 5 and D7 where *n* = 6 (\**p* = 0.0164, \*\*\**p* = 0.0007). **g** Quantification of soluble Plet1 in BALF of D7 IAV-infected *Ccr2*-/- mice, adoptively transferred at D3 p.i. with BMDM or TR-AM subsets flow-sorted from BALF of IAV-infected WT mice at the indicated time points (*n* = 3) (\*\**p* = 0.0087, \*\*\**p* = 0.0001, \*\*\*\**p* < 0.0001). **h** Plet1 concentration in supernatant of non-infected or infected AEC (24 h) (*n* = 5); in supernatant of BALF-isolated TR-AM treated for 12 h with conditioned medium of non-infected (*n* = 4) or infected AEC (12 h, *n* = 6, and 24 h, *n* = 5) or of untreated TR-AM (*n* = 6) (\*\*\**p* < 0.0001). Data are representative of three independent experiments shown as mean ± SEM statistical differences were calculated using one-way ANOVA and Tukey's post-hoc tests. BMDM bone marrow-derived macrophage, AEC alveolar epithelial cell, BALF bronchoalveolar lavage fluid, TR-AM tissue-resident alveolar macrophage, D day. Source data are provided as a Source Data file.

data indicate that, BMDM2-expressed Plet1 drives AEC proliferation, resulting in alveolarization in lung organoids, and that Plet1 restores alveolar epithelial barrier function after infectious challenge ex vivo, effects that we suggest to be mediated via MAPK activation and Src kinase inhibition, respectively.

## BMDM2-expressed Plet1 drives alveolar epithelial progenitor cell (AEC II) expansion and protection of lung barrier function in IAV-induced lung injury in vivo

We next confirmed that Plet1 is a crucial mediator of the macrophage epithelial cell protective phenotype in vivo, focusing on BMDM2 as the

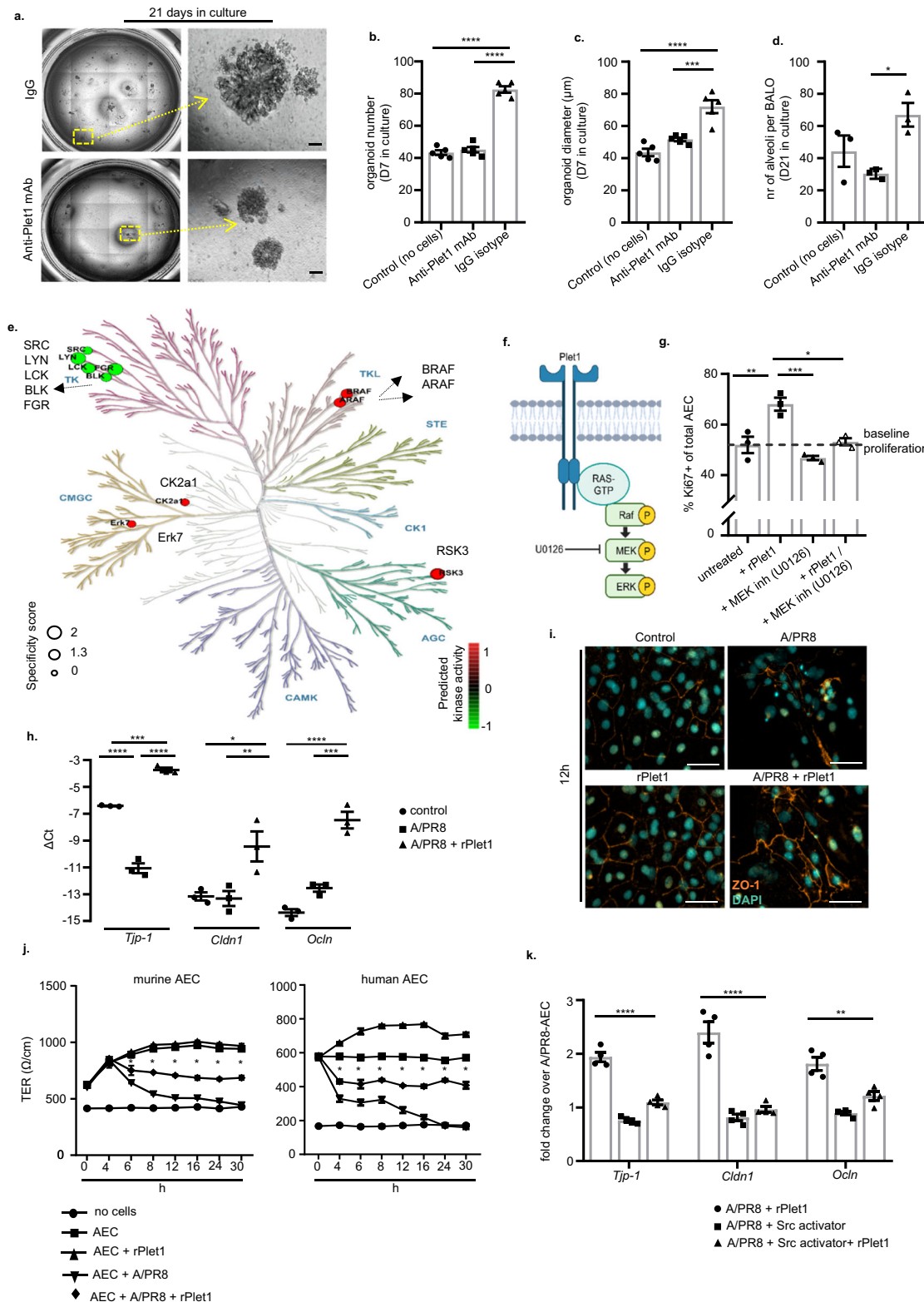

key transitional cell population upregulating Plet1 during the course of IAV infection. We applied the BMDM2 → $Ccr2^{-/-}$ mice intrapulmonary transfer model using BMDM2 from $Plet1^{-/-}$ mice or BMDM2 pre-incubated with anti-Plet1 mAb, and analyzed AEC parameters at D7 where we observed the peak of alveolar injury. $Plet1^{-/-}$ mice were obtained by generation of $Plet1^{\text{tdtomato-flox/flox}}$ (termed $Plet1^{\text{flox/flox}}$) crossbred with Cre-deleter mice, with loss of $Plet1$ mRNA and concomitant loss of tdTomato reporter confirmed in BMDM2 of IAV-

infected $Plet1^{-/-}$ mice compared to Cre-negative $Plet1^{\text{flox/flox}}$ mice (Supplementary Fig. 6a–c). Plet1 neutralization or knockout in BMDM2 reduced the beneficial effects of BMDM2, resulting in increased AEC apoptosis, reduced expression of tight junction genes in epithelial cells, and alveolar barrier dysfunction, as well as reduced proliferation of AEC II, as compared to controls (Supplementary Fig. 6d–g). Finally, $Ccr2^{-/-}$ mice receiving $Plet1$-deficient or Plet1-neutralized BMDM2 revealed persistent tissue inflammation compared to controls at D7

**Fig. 5 | Soluble Plet1 drives AEC expansion and promotes alveolar epithelial barrier function in ex vivo models. a** Representative image of BALO co-cultured with BMDM2 for 21D and treated with anti-Plet1 mAb or isotype control. Scale bar represents 50 μm. **b** Quantification of BALO numbers; single data points represent the mean of organoid numbers per each well ($n = 5$) (****$p < 0.0001$). **c** Mean diameter (μm) of organoids ($n = 40–87$) per each well, $n = 5$ well/group, after 7D in culture; single data points represent mean diameter of organoids per each well (****$p < 0.0001$, ***$p = 0.0006$). **d** average numbers of alveoli per BALO after 21D in culture (control) or co-culture with BMDM2 and anti-Plet1 mAb or isotype control (alveoli were counted from $n = 3$ BALO per group) (*$p = 0.0252$). **e** Predicted kinase activity (5 top phosphorylated, red, and de-phosphorylated kinases, green; shown within phylogenetic kinome tree) in AEC treated with 20 ng/ml rPlet-1 for 12 h. **f** Suggested kinase signaling pathway activated by Plet1. The figure was created with BioRender. **g** Assessment of Ki67 staining in murine primary AEC culture treated with rPlet1 (20 ng/ml) and/or MEK inhibitor U0126 (10 μM) ($n = 3$) (*$p = 0.0065$, **$p = 0.004$, ***$p = 0.0006$). **h** Quantification of tight junction gene expression (*Tjp-1, Cldn1, Ocln*) by qPCR in murine AEC IAV-infected and treated with rPlet1 (20 ng/ml) ($n = 3$) (*Tjp-1* ***$p = 0.0004$, ****$p < 0.0001$; *Cldn1* *$p = 0.0282$,

**$p = 0.0236$; *Ocln* ***$p = 0.0003$, ****$p < 0.0001$). **i** Representative image of ZO-1 localization in AEC monolayers, IAV-infected and/or treated with rPlet1. Scale bar represents 50 μm. **j** Time course of transepithelial resistance (TER) in murine ($n = 3$) and human AEC ($n = 4$) monolayers on transwells, IAV-infected and/or treated with rPlet1 (*$p < 0.0001$ for the comparison between AECs+A/PR8 and AECs+A/PR8+rPlet1). **k** Quantification of tight junction gene expression (*Tjp-1, Cldn1, Ocln*) by qPCR in murine AEC IAV-infected and treated with rPlet (20 ng/ml) with/without Src activator (10 μM). Results are shown as fold change over IAV-infected AEC without rPlet1 ($n = 4$) (*Tjp-1* ****$p < 0.0001$; *Cldn1* ****$p < 0.0001$; *Ocln* **$p = 0.0024$). Data are representative of three independent experiments showing mean ± SEM calculated using one-way ANOVA and Tukey's post-hoc tests. **j** represents a single experiment performed with 3 (mouse) or 4 (human) different biological independent samples showing mean ± SEM and results were analyzed by two-way ANOVA and Tukey's post-hoc test. BMDM bone marrow-derived macrophage, AEC alveolar epithelial cell, BALO bronchoalveolar lung organoid, TER transepithelial resistance, TR-AM tissue-resident alveolar macrophage, D day, inh inhibitor. Source data are provided as a Source Data file.

(Supplementary Fig. 6h). We next generated a conditional (Tamoxifen (Txf)-dependent) BMDM-specific *Plet1*-knockout line by breeding *Plet1*flox/flox mice to *Cx3cr1*tm2.1(cre/ERT2)Jung mice (*Cx3cr1*iCre-*Plet1*flx/flx). Circulating classical monocytes express high levels of Cx3cr1, while TR-AM do not[41,42]. Of note, if Cx3cr1+ monocytes give rise to BMDM and then to TR-AM, these will permanently remain *Plet1*-deficient and tdTomatoneg after Cre recombination by Txf administration, even after the expression of endogenous *Cx3cr1* has ceased during differentiation into TR-AM[43], allowing genetic targeting of BMDM and BMDM-derived TR-AM. In line, Cx3cr1neg (not BMDM-replenished, fetal monocyte-derived) TR-AM of Txf-treated *Cx3cr1*iCre-*Plet1*flx/flx mice revealed similar levels of *Plet1* mRNA and tdTomato expression as those of Txf-treated *Plet1*flx/flx control mice, whereas BMDM2 (analyzed at D21 p.i.) showed significantly reduced *Plet1* mRNA and tdTomato expression (Supplementary Fig. 6i–l; tdTomato-MFI of C57BL/6 WT TR-AM and D21 BMDM2 depicted for comparison).

We next co-cultured *Cx3cr1*iCre-*Plet1*flx/flx versus *Plet1*flx/flx BMDM2 flow-sorted at D21 from IAV-infected, Txf-treated mice with BALO as described in Fig. 3c. As expected, loss of Plet1 in BMDM2 resulted in reduced organoid numbers and growth at D7 and D21 of culture, with reduced BALO complexity and limited alveolarization at D21 (Fig. 6a–d). We next analyzed parameters of persistence of alveolar injury and induction of alveolar repair in Txf-treated *Cx3cr1*iCre-*Plet1*flx/flx versus *Plet1*flx/flx mice, a process starting around D10 post IAV infection. Lung histopathology reveals that *Plet1*flx/flx mice had widely (D10) or completely (D14, D21) resolved the injury-associated inflammation, whereas dense inflammatory infiltrates were found in *Cx3cr1*iCre-*Plet1*flx/flx mice with partial resolution until D21 (Fig. 6e). Concomitantly, AEC apoptosis was still significantly increased and AEC II proliferation decreased in *Cx3cr1*iCre-*Plet1*flx/flx mice at D10 p.i., together with reduced expression of tight junction genes (*Tjp-1, Cldn1, Ocln*), resulting in higher alveolar barrier leak persisting until D14 p.i. (Fig. 6f–i). Alveolar repair was incomplete in *Cx3cr1*iCre-*Plet1*flx/flx mice with evidence of disrupted alveoli (Fig. 6j, asterisks, Fig. 6k) and increased thickness of septa (Fig. 6j). Finally, after IAV infection with a virus dose that resulted in low mortality in control mice (Fig. 6l), *Cx3cr1*iCre-*Plet1*flx/flx mice revealed significantly reduced survival between D10 and D14 p.i., coinciding with BMDM2 appearance in the interstitium and airspaces that confer protection in the *Plet1*flx/flx control mice (according to scRNAseq and FACS data; Figs. 1e and 2c and Supplementary Fig. 2c). Plet1 was not detected in BALF of Txf-treated *Cx3cr1*iCre-*Plet1*flx/flx mice compared to *Plet1*flx/flx control mice at D7 and 21 p.i., indicating that BMDM2 (and BMDM2-derived TR-AM) are a major source of Plet1 in the alveolar compartment (Supplementary Fig. 6m, q). Of note, quantification of cytokines and chemokines, and extent of inflammatory infiltrates revealed that lack of BMDM2-Plet1 delayed the resolution of the inflammatory responses at d10-14, whereas the level of

tissue fibrosis was not affected (Supplementary Fig. 6n–p) at the indicated time points. These data indicate that Plet1 expressed in BMDM2 (and likely, in BMDM-derived TR-AM arising after D14) is crucial for lung tissue recovery and to survive severe viral infection.

## Orotracheal administration of rPlet1 rescues mice after lethal IAV infection with therapeutic implications

Treatment of human IAV-infected AEC with rPLET1 resulted in significant improvement of barrier function (Fig. 5j). To verify the putative relevance of PLET1 in the context of human disease, we quantified PLET1 levels in BALF of patients with IAV-induced ARDS compared to a control group of patients with non-inflammatory/non-infectious lung disease (patient characteristics provided in Supplementary Table), and detected high levels in a subset of IAV-ARDS patients (Fig. 7a). To correlate PLET1 levels in IAV-ARDS BALF with the level of alveolar epithelial injury, we additionally quantified total BALF protein as a measure of alveolar barrier dysfunction in both cohorts (Fig. 7b), and PLET1 concentrations negatively correlated with total BALF protein concentrations (Fig. 7c), indicating that soluble PLET1 was present in BALF of patients with IAV-ARDS and might exert similar barrier-protective functions as observed in mice. We next evaluated the therapeutic potential of alveolar administration of rPlet1 in mice at D3 p.i., using an experimental setup (Fig. 7d) aimed to attenuate the peak of lung injury at D7 to D10[8] in a severe IAV infection model. rPlet1 treatment reduced the injury-associated tissue inflammation, reduced AEC apoptosis, upregulated expression of tight junction genes *Tjp-1* and *Cldn1* together with improvement of alveolar barrier function, and induced substantial AEC II proliferation, indicative of a tissue-protective and -regenerative effect (Fig. 7e–i). Of note, quantification of cytokines (IL-6, KC, MIP1α, IL10), extent of inflammatory infiltrates and collagen deposition revealed that rPlet1 did not affect inflammatory responses or levels of tissue fibrosis (Supplementary Fig. 7a–c). In addition, viral clearance was not affected by rPlet1 treatment either (Supplementary Fig. 7d). Finally, mice challenged with a lethal dose of IAV were rescued by 85.7% by rPlet1 compared to control treatment (Fig. 7j), suggesting that local administration of rPlet1 may represent a putative treatment strategy in human virus-induced ARDS.

## Discussion

Respiratory infections by endemic and emerging viruses pose a major threat to human health as currently evidenced by the COVID-19 pandemic. Lung macrophages have been attributed a crucial role in driving the severity of virus-induced lung tissue injury in IAV[8,11,44] and COVID-19[10,45]. Several studies revealed that TR-AM pools are depleted upon viral pneumonia to different extent, depending on viral strain and dose, and are gradually replaced by BMDM during the healing phase, resulting in reprogramming of the TR-AM pool that can either

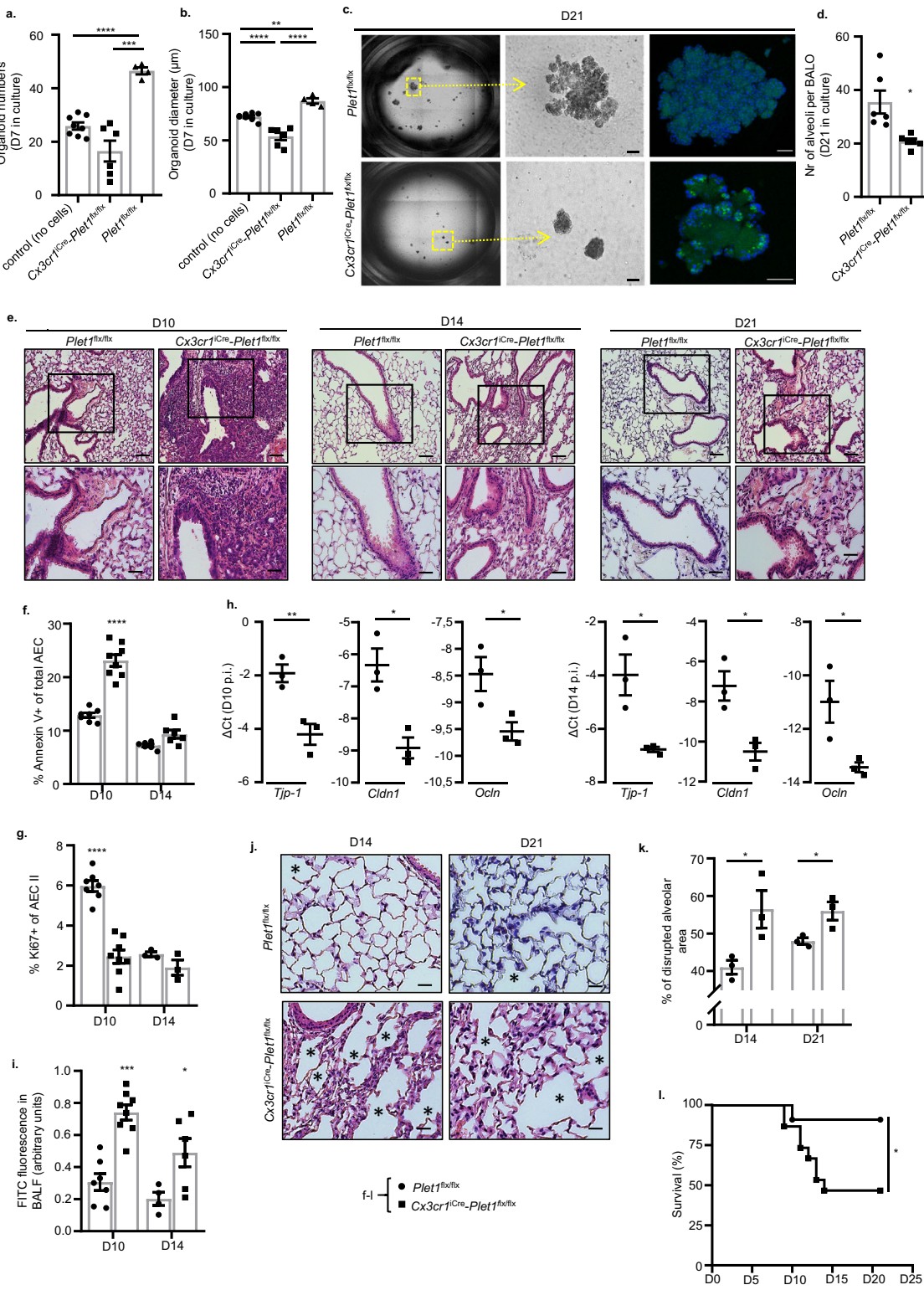

confer trained immunity, tolerance or resilience to subsequent challenges[1–3,46–48]. The bone marrow origin of replaced, rather than training of locally persisting TR-AM by pathogen encounter or microenvironmental cues, seem to determine the net functional phenotype of the TR-AM pool at least after viral infections. To date, most of the data relating to macrophage ontology and the resulting phenotype have focused on their contribution to subsequent infections and remain conflicting; however, far less is known about their role in the resolution of inflammation and tissue repair[3,4,49–51].

Our data provide evidence that a specific epithelial repair program is induced in alveolar macrophages, via a distinct trajectory of functional BMDM specification from pro-inflammatory/injury-promoting to tissue-healing, with the latter effects mediated by Plet1. The BMDM2 Plet1-driven epithelial-protective phenotype remained upregulated in replenished, experienced TR-AM for at least 60D p.i., whereas the direct epithelial-proliferative effect declined at 35D p.i. As we could detect Plet1 expression in TR-AM at D60 p.i., other co-factors may be needed in addition to exert the pro-proliferative effect of Plet1

**Fig. 6 | BMDM2-expressed Plet1 drives epithelial progenitor (AEC II) cell expansion and improves lung barrier function in IAV-induced lung injury in vivo. a** Quantification of BALO numbers after 7D in culture (control) ($n = 8$) or co-culture with BMDM2 derived from Txf-treated control ($Plet1^{flx/flx}$) ($n = 4$) or $Cx3cr1^{iCre}$-$Plet1^{flx/flx}$ ($n = 6$) mice; single data points represent mean of organoid numbers per each well (***$p = 0.0008$, ****$p < 0.0001$). **b** BALO diameter (μm) from the same experiment; single data points represent mean diameter of organoids ($n = 20$–$49$) per each well. Groups: Control (no cells) ($n = 8$ well); $Plet1^{flx/flx}$ ($n = 4$ well) and $Cx3cr1^{iCre}$-$Plet1^{flx/flx}$ ($n = 6$ well) (**$p = 0.0026$, ****$p < 0.0001$). **c** Representative image of BALO at 21D of culture, stained with LysoTracker® Green DND. The scale bar in the EVOS FL microscope images (middle) represents 50 μm while in confocal images 25 μm (top) and 50 μm (bottom). **d** Average numbers of alveoli per BALO from the experiment in (**c**); quantified from $n = 6$ BALO/group (*$p = 0.0182$). **e** Lung histological sections of IAV-infected, Txf-treated $Cx3cr1^{iCre}$-$Plet1^{flx/flx}$ or $Plet1^{flx/flx}$ mice obtained after 10/14/21D p.i., stained with H&E. Images underneath are magnifications of the areas within squares. Scale bars represent 100 μm (top) and 50 μm (bottom). **f** Percentage of Annexin $V^+$ AECs in $Plet1^{flx/flx}$ mice at day 10 ($n = 7$) and day 14 ($n = 6$) or $Cx3cr1^{iCre}$-$Plet1^{flx/flx}$ mice at day 10 ($n = 8$) and day 14 ($n = 6$) (****$p < 0.0001$). **g** Percentage of Ki67$^+$ AECs in $Plet1^{flx/flx}$ mice at day 10 ($n = 7$) and day 14 ($n = 3$) or $Cx3cr1^{iCre}$-$Plet1^{flx/flx}$ mice at day 10 ($n = 8$) and day 14 ($n = 3$) (****$p < 0.0001$). **h** mRNA expression (qPCR) of tight junction component genes in the lung ($n = 3$) ($Tjp-1$ D10 **$p = 0.01$, D14 *$p = 0.0224$; $Cldn1$ D10 *$p = 0.013$,

D14 *$p = 0.0185$; $Ocln$ D10 *$p = 0.0405$, D14 *$p = 0.0348$). **i** Quantification of barrier dysfunction by FITC-Albumin fluorescence analysis in BALF of $Plet1^{flx/flx}$ mice at day 10 ($n = 7$) and day 14 ($n = 4$) or $Cx3cr1^{iCre}$-$Plet1^{flx/flx}$ mice at day 10 ($n = 8$) (****$p < 0.0001$) and day 14 ($n = 6$) (*$p = 0.0368$). **f**–**i** refer to the experiment described in (**e**). **j** Lung sections of IAV-infected $Plet1^{flx/flx}$ or $Cx3cr1^{iCre}$-$Plet1^{flx/flx}$ mice obtained at D14/21 p.i., stained with H&E. Scale bars represent 50 μm. Images provide a detailed visualization of specific regions from the corresponding images in 6e (bottom) while maintaining the same magnification as in 6e, bottom. * indicates areas of disrupted, non-repaired alveoli. **k** Quantification of disrupted alveolar area in lung slides of IAV-infected mice. Single data points represent the mean of three randomly chosen areas from each paraffin lung section ($n = 3$ biologically independent mouse lung sections for each group) (D14 *$p = 0.0439$, D21 *$p = 0.0367$). **l** Survival analysis of IAV-infected, Txf-treated $Cx3cr1^{iCre}$-$Plet1^{flx/flx}$ ($n = 15$) or $Plet1^{flx/flx}$ mice ($n = 11$) (*$p = 0.029$). Data are representative of three independent experiments; bar graphs show means ± SEM and single data points. Significance was calculated using Brown Forsythe and Welch ANOVA followed by Games−Howell's test on (**a**) and ANOVA followed by Tukey's post-hoc tests in (**b**), two-sided Student's t-test performed in (**d**), (Welch's correction) (**f**) (Welch's correction) (**g**, **h**, **i**, **k**) and Log-rank (Mantel−Cox) test in (**l**). AEC alveolar epithelial cell, BALO bronchoalveolar lung organoid, BALF bronchoalveolar lavage fluid, D day. Source data are provided as a Source Data file.

at later stages. Also, despite our RNA velocity data on a likely in vivo transition of BMDM1 to BMDM2 phenotype, we cannot exclude that an additional wave of BMDM2 enters the lungs at later stages of the infection course, when repair of the lungs starts to be initiated.

Plet1 is a GPI-anchored membrane protein, mediating epithelial repair responses such as keratinocyte migration in wound healing[20] and proliferation of Lgr5$^+$ colonic stem cells after injury[21] in a cell-autonomous manner. It was recently identified as a marker gene for alveolar macrophage specification among tissue-resident macrophages of various organs[38]; however, no data so far have revealed a role for Plet1 in myeloid cell-mediated tissue repair. At least two different effects were induced by Plet1 in the lung epithelium ex vivo, in organoids, and in vivo, ultimately resulting in the protection of barrier function: First, Plet1 mediated organoid outgrowth from BASC and subsequent alveolarization in BALO, and increased proliferation of AEC II in vivo, indicating its pro-proliferative action on lung epithelial progenitor cells. Furthermore, Plet1 protected AEC from apoptosis, an effect likely associated with the self-renewal program induced[52]. Proliferation of progenitor cells in different niches of the distal lung is a crucial step in repair after viral lung injury. The alveolar epithelium can be re-established from different local stem/progenitor pools, the most important being Axin2$^+$ subsets of AEC II, that via Krt8$^+$ intermediates de-differentiate into AEC I to ultimately re-establish alveolar epithelial barrier function. At least in mice, alveolar repair after IAV injury is additionally driven by expansion of bronchoalveolar stem cells, the BALO cells of origin, that de-differentiate into AEC II and further into AEC I[36,53,54]. In search of a putative signaling mechanism by which Plet1 would drive proliferation of alveolar epithelial cells with regenerative capacity, we identified a MAPK pathway, as MEK inhibition completely abolished Plet1-induced proliferation, suggesting that Raf-MEK-ERK signaling was involved. Similar findings were reported[21], where Plet1 induced colonic stem cell proliferation via ERK1/2. However, we cannot exclude further mechanisms being involved, e.g. CK2a1 (a further hit in the phosphokinome screen) known to mediate cell survival and proliferation[55], or others. Ongoing experiments aimed at identification of the Plet1 receptor will likely reveal the full spectrum of Plet1-driven progenitor cell activation.

A second effect elicited by Plet1 was re-establishment of epithelial barrier properties, associated with increased expression of tight junction-associated molecules ZO-1, occludin and claudin-1, the latter previously described to confer sealing properties of tight junctions in airway epithelia[56]. Disruption of the apical junction complex in pulmonary epithelial cells is one of the pathomechanisms driving loss of

barrier integrity in IAV infection[57]. Therefore, we expected that, in addition to the regenerative program elicited in lung epithelial progenitor cells, induction of tight junction proteins by Plet1 is crucial in both, protection from IAV-induced epithelial barrier disruption, and re-sealing of newly generated AEC. We speculated that this re-sealing program was mediated by pathways distinct from those driving proliferation as it was demonstrated that inhibition of Src kinases prevented inflammation-dependent loss of tight junction proteins in lung epithelial cells[58]. In addition to our own phosphoproteome prediction data showing that Plet1 treatment was associated with reduction of Src kinase activity, treatment with a Src kinase activator abolished these Plet1-induced tight junction protein upregulation. This certainly does not rule out further mechanisms involved, but indicates that, likely, two separate regeneration and repair programs are induced by Plet1 in alveolar epithelial cells, driving proliferation and re-establishment of cell-cell contacts, respectively.

The mechanism by which macrophage Plet1 interacts with the epithelium is an important question to be discussed. We speculate that it might act as a soluble protein, at least in addition to its membrane-expressed form, for several reasons. First, use of recombinant soluble Plet1 instead of Plet1$^+$ macrophages reproduced many of the observed effects in vitro and in vivo. In fact, Plet1 can be removed from cells by the addition of Phospholipase C, suggesting that shedding from macrophage surfaces might occur in vivo[20]. In addition, shedding of Plet1 from the surface of TR-AM occurred in in vitro culture and was significantly increased upon treatment with conditioned medium of infected AEC, indicating that mediators released from IAV-injured AEC, possibly cytokines or danger-associated molecular patterns (DAMPS), were a major signal to increase soluble Plet1 concentrations at sites of infection or injury. Interestingly, naive, non-replenished TR-AM express Plet1, but did not release high amounts upon transfer into inflamed/injured lungs of IAV-infected $Ccr2^{-/-}$ mice, whereas transfer of BMDM2 or experienced TR-AM resulted in highly increased Plet1 BALF concentrations. This could explain why BMDM2/experienced TR-AM transfer, but not naïve TR-AM transfer, protected the lung epithelium of IAV-infected $Ccr2^{-/-}$ mice.

TR-AM self-renewal has been pointed out as a major contributor to TR-AM replenishment in lung viral infection, being progressively outcompeted by BMDM-derived TR-AM following IAV infection[3,59]. Although we cannot fully exclude a contribution of self-renewing fetal monocyte-derived TR-AM to Plet1-mediated repair, this seems unlikely since $Cx3cr1^{iCre}$-$Plet1^{flx/flx}$ mice had undetectable levels of Plet1 in BALF. With regard to this transgenic model, the possibility exists

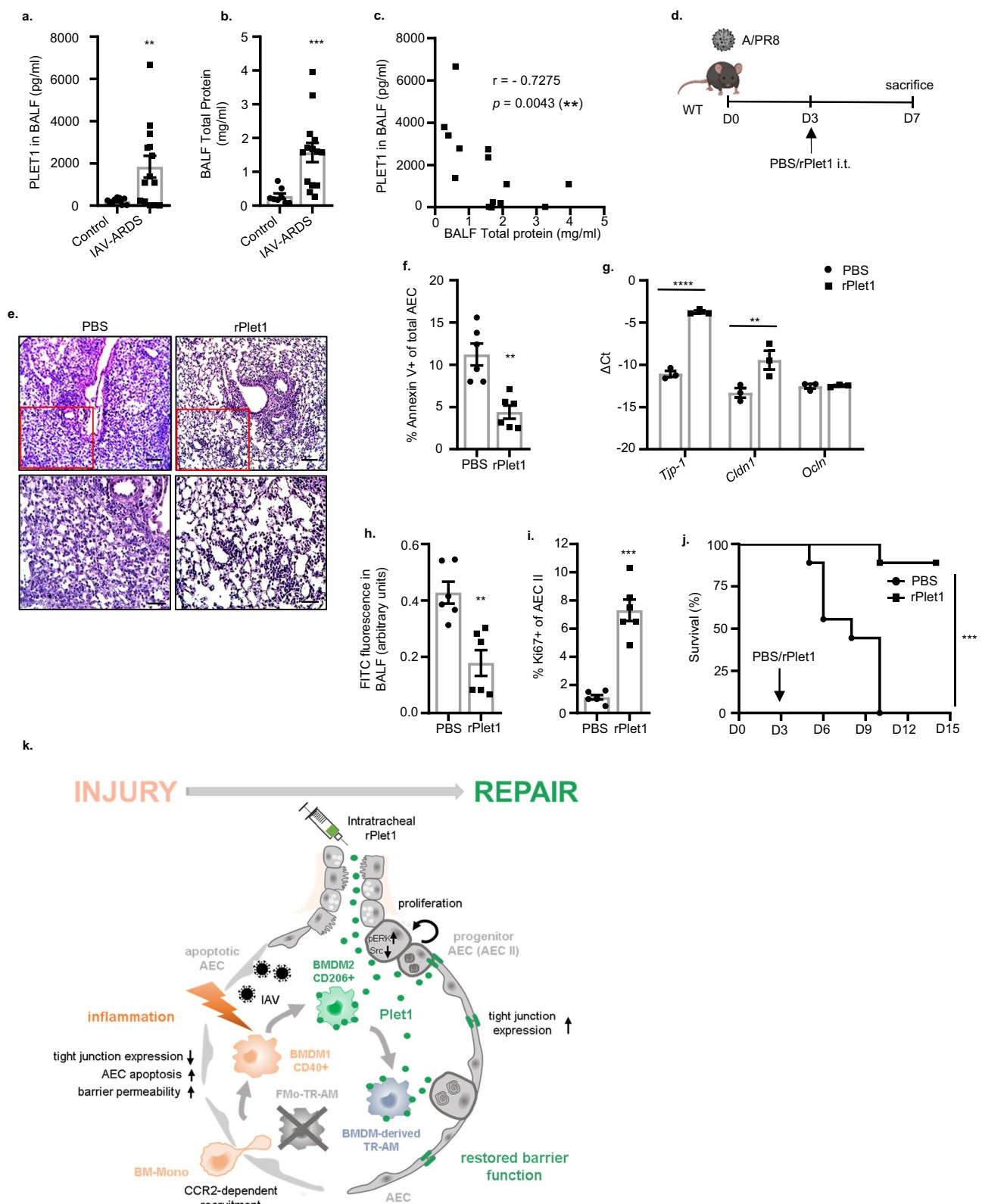

that Plet1 knockout in other Cx3cr1-expressing cells might contribute to the outcome of *Cx3cr1*[iCre]-*Plet1*[flx/flx] mice. For instance, interstitial macrophages, DCs and T cells can express *Cx3cr1*; however, we found no significant levels of Plet1 expression in those cell populations in IAV-infected mice by flow cytometry (Supplementary Fig. 6q). Together, our data reveal an innovative mechanism of macrophage-mediated epithelial repair that is key to survive IAV-induced lung injury (Fig. 7k). Independently of the source; it has to be highlighted that local administration of recombinant Plet1 reproduced the epithelial-protective and -regenerative effects driven by Plet1+ macrophage populations, and rescued 85% of mice from fatal influenza. Remarkably, in BALF of a small cohort of patients with IAV-induced ARDS, PLET1 concentrations negatively correlated with a marker of alveolar injury, underscoring Plet1 administration as a putative therapeutic approach in patients with virus-induced lung injury and beyond, that are urgently needed to date.

**Fig. 7 | PLET1 is present in patients with human IAV-induced ARDS and rPlet1 administration protects lung epithelial barrier function and rescues mice after lethal IAV infection. a** Soluble PLET1 and **b** total protein concentration in BALF of patients with IAV-induced ARDS ($n = 14$) or healthy patients ($n = 8$) (patient characteristics provided in Supplementary Table) (**$p = 0.0075$, ***$p = 0.0005$). **c** PLET1 concentration was negatively correlated with total protein concentration ($n = 14$). **d** Schematic representation of the experiments conducted in (**e**)–(**i**), rPlet1 (5 μg; squares) or PBS (circles) were applied intratracheally (i.t.) at D3 p.i. The figure was created with BioRender. **e** Lung histological sections of IAV-infected mice stained with H&E at D7 p.i. Bottom images underneath are magnifications of the top images (red squares); scale bars represent 100 μm (top) and 50 μm (bottom). **f** Percentage of Annexin V⁺ AECs ($n = 6$) (**$p = 0.0011$). **g** mRNA expression (qPCR) of tight junction component genes in flow-sorted EpCam⁺ cells ($n = 3$) (*$p = 0.0046$, ****$p < 0.0001$). **h** Quantification of barrier dysfunction by FITC-Albumin fluorescence analysis in BALF ($n = 6$) (**$p = 0.002$). **i** Ki67⁺ percentage of AEC II indicating proliferation ($n = 6$) (***$p = 0.0003$). **j** Survival of IAV-infected mice ($5 \times 10^2$ pfu) treated with PBS or rPlet1 at D3 p.i. ($n = 9$/group in two separate experiments) (***$p = 0.0001$). **k** Schematic summary: CD206⁺ BMDM2- and BMDM2-derived TR-AM express Plet1 driving AEC II proliferation and upregulation of tight junction genes, ultimately resulting in AEC barrier repair. Application of exogenous, recombinant Plet1 reproduces these effects, highlighting its therapeutic potential. The figure was created with BioRender. Data are representative of three independent experiments showing mean ± SEM. Significance was calculated using two-sided Student's t-test on (**a**) (Welch's correction), (**b**) (Welch's correction), (**f, h, i**) (Welch's correction), Spearman's rank correlation test in (**c**), Log-rank (Mantel–Cox) test in (**j**), (**g**) two-way ANOVA (followed by Šídák's test). AEC alveolar epithelial cell, BALF bronchoalveolar lavage fluid, D day. Panel (**k**) was generated by BioRender. Source data are provided as a Source Data file.

## Methods

### Study approvals
Animal experiments were approved by the regional authorities of the State of Hesse (Regierungspräsidium Giessen) and by the Institutional Ethics Committee at the IBioBA Institute. Human lung tissue and BALF samples were obtained from patients who underwent lobectomy or received bronchoscopy for diagnostic reasons after written consent. Use of human lung tissue and BALF samples was approved by the University of Giessen Ethics Committee.

### Mice
C57BL/6 and B6.SJL-*Ptprc*ᵃ mice expressing the CD45.1 alloantigen (Ly5.1 PTP) on circulating leukocytes with C57BL/6 genetic background were purchased from Charles River Laboratories. *Ccr2*⁻/⁻ (B6.129P2-*Ccr2*ᵗᵐ¹ᴹᵃᵉ) mice were generated as described previously and backcrossed to the C57BL/6 background[60]. All mice together with their respective littermate controls were bred under specific pathogen-free conditions and were used between 10 and 14 weeks of age.

Housing conditions for the mice are as follows: light/dark cycle of 4/10 h, temperature maintained at 22 % 2 °C, relative humidity at 55 % 10%.

### Generation of Plet1 transgenic mice
Mice were generated in cooperation with Taconic Biosciences with a tdTomato (tandem dimer tomato red) reporter gene tandemly expressed with a floxed *Plet1* gene on C57BL/6 background to obtain constitutive knock-in of T2A-tdTomato in the *Plet1* locus (Plet1 reporter) with optional conditional knockout of the *Plet1* gene together with the reporter gene. The sequences for the T2A and the open reading frame of tdTomato have been inserted between the last amino acid and the translation termination codon in exon 4, and exon 3 has been flanked by loxP sites. The targeting vector was generated using BAC clones from the C57BL/6J RPCI-23 BAC library and was transfected into C57BL/6N Tac ES cells. The presence of the T2A sequence resulted in the co-translational cleavage between the PLET1 and tdTomato proteins, with co-expression of the Plet1 and tdTomato proteins under the control of the endogenous *Plet1* promoter. Deletion of exon 3 results in the loss of function of the *Plet1* gene by generating a frameshift from exon 2 to exon 4 (premature Stop codon in exon 4). Homozygous *Plet1*ᵗᵈᵗᵒᵐᵃᵗᵒ⁻ᶠˡˣ/ᶠˡˣ (abbreviated *Plet1*ᶠˡˣ/ᶠˡˣ) mice. This line was further crossbred with B6.129P2(C)-*Cx3cr1*ᵗᵐ²·¹⁽ᶜʳᵉ/ᴱᴿᵀ²⁾ᴶᵘⁿᵍ mice (Jackson) for the generation of tamoxifen-responsive Cx3cr1⁺ (BMDM-specific) *Plet1* conditional knockout mice (abbreviated *Cx3cr1*ⁱᶜʳᵉ-*Plet1*ᶠˡˣ/ᶠˡˣ) by tamoxifen (TXF) feeding. Tamoxifen was administered via chow (Taconic Biosciences, GmbH, 0.4 g/kg) prior to the start of the experiment to target circulating Cx3cr1+ BMDM precursors (monocytes) before recruitment to the lung (including *Plet1*ᶠˡˣ/ᶠˡˣ controls when compared to *Cx3cr1*ⁱᶜʳᵉ-*Plet1*ᶠˡˣ/ᶠˡˣ mice, to control for food-/Txf-caused effects). Breeding of *Plet1*ᶠˡᵒˣ/ᶠˡᵒˣ with Cre-deleter (B6.C-Tg (CMV-cre)1Cgn/J) mice resulted in ubiquitous, non-conditional loss of the *Plet1* gene (abbreviated *Plet1*⁻/⁻).

### In vivo infection and treatment protocols
Mice were anesthetized and oro-tracheally inoculated with 2.5-5×10² pfu/ml (Plaque-forming units per milliliter) of influenza A virus PR8 (A/PR/8/1934(H1N1)) diluted in 70 μl sterile PBS⁻/⁻. In the treatment approach, 5 μg recombinant murine Plet1 (CUSABIO) dissolved in 70 μl sterile PBS⁻/⁻ or PBS⁻/⁻ alone was applied oro-tracheally to infected mice on D3 p.i. Treated mice were monitored 1–3 × per day and weighed daily. In survival experiments mice were euthanized when reaching a defined illness score validated for A/PR8 infection and predicting death of the animal within short time. The score included several morbidity criteria (ruffled fur, breathing, movement), and high body weight loss, according to the legal requirements. Euthanized mice were noted as dead in the survival analyses.

Bone marrow (BM) chimeric mice were generated as follows: BM cells were isolated under sterile conditions from the tibias and femurs of wt C57BL/6 donor mice (expressing the CD45.2 alloantigen) as previously described and transferred to CD45.1 alloantigen-expressing recipient mice which had received total body irradiation (6 Gy)[30]. To assess BM engraftment, the proportion of donor CD45.2-expressing leukocytes in blood, BALF and lung homogenate was analyzed by flow cytometry (FACS). Two weeks after transplantation, regularly >95% of circulating leukocytes were of donor type (CD45.2), whereas >95% TR-AM were of recipient type (CD45.1). Chimeric mice were housed under specific pathogen-free conditions for 14 days before PR8 infection.

For pulmonary transfer of macrophages, BMDM or TR-AM were obtained by bronchoalveolar lavage at the indicated time points p.i. from PR8-infected mice of the given genotype. TR-AM were identified by the signature CD45⁺Ly6CˡᵒLy6G⁻CD11bˡᵒCD11cʰⁱSiglecFʰⁱ, BMDM1 and BMDM2 were identified by the signature CD40ʰⁱCD206ˡᵒCD45⁺Ly6C⁺Ly6G⁻CD11bʰⁱCD11cˡᵒSiglecF⁻ and CD40ˡᵒCD206ʰⁱCD45⁺Ly6C⁺Ly6G⁻CD11bʰⁱCD11cˡᵒSiglecF⁻, respectively, and flow-sorted. Cell suspensions were used when the purity was ≥95% as ensured by FACS and pappenheim-stained cytospins. 50,000 cells in 50 μl sterile PBS⁻/⁻ were transferred oro-tracheally to PR8-infected *Ccr2*⁻/⁻ mice on D3 p.i. For Plet1 neutralization on BMDM2, cells were incubated for 10 min at 4 °C with anti-Plet1 antibody (R&D Systems) or recommended IgG isotype control, washed, and transferred directly thereafter.

### Flow cytometry and cell sorting
Cells ($1-5 \times 10^5$) derived from bronchoalveolar lavage or from lavaged, perfused, and homogenized murine lungs were resuspended in FACS buffer (PBS⁻/⁻, 10% FBS, 0.1% NaN3), preincubated with Fc block for 5 min and stained with fluorochrome-conjugated antibodies for 30 min at 4 °C. For Annexin V staining, cells were resuspended in Annexin V staining buffer (10 mM HEPES, 140 mM

NaCl, and 2.5 mM $CaCl_2$). The fluorochrome-labeled antibodies used for FACS analysis of macrophage subsets/phenotypes, Plet1 expression, AEC proliferation and apoptosis are CD326 (EpCam) (APC-Cy7, clone G8.8, Cat No 118218 BioLegend) (1/100), CD24 (PE-Cy7, clone M1/69, BioLegend, Cat No 101821) (1/100), CD31 (Alexa Fluor 488 clone MEC13.3, BioLegend, Cat No 102513, or pacific blue clone 390, BioLegend, Cat No 102413) (1/50), CD45 (FITC Clone 30-F11 BioLegend, Cat No 103107, APC-Cy7 30-F11 BioLegend, Cat No 103115, or pacific blue, clone 30-F11 BioLegend, Cat No 103125) (1/100), T1α/ podoplanin (APC, clone 8.1.1 BioLegend, Cat No 127409) (1/20), Ki67 (FITC clone 16A8 BioLegend, Cat No 652409 or BV605, clone 16A8, BioLegend, Cat No 652413; PE, clone 16A8, BioLegend, Cat No 652403) (1/50), and corresponding isotype control FITC Rat IgG2a, κ Isotype Ctrl Antibody (clone RTK2758, BioLegend, Cat No 400505, 1/50), Brilliant Violet 605™ Rat IgG2a, κ Isotype Ctrl Antibody (clone RTK2758, BioLegend, Cat No 400539, 1/50), PE Rat IgG2a, κ Isotype Ctrl Antibody (clone RTK2758, BioLegend, Cat No 400508, 1/50), Annexin V (Alexa Fluor 647, clone 640907 Invitrogen, Cat No A23204, PE, clone 640907, Invitrogen, Cat No A35111) (1/20), GR-1 (PE-Cy7 clone RB6-8C5, BioLegend, Cat No 108415, or PerCP, clone RB6-8C5, BioLegend, Cat No 108425) (1/100), Ly6G (APC clone 1A8 BioLegend, Cat No 127617, or PE-Cy7, clone 1A8, BioLegend, Cat No 127613) (1/50), Ly6C (FITC, clone AL-21 BD Biosciences, Cat No 553104) (1/20), Siglec-F (PE clone E50-2440, BD Biosciences, Cat No 562068 or BV421, clone E50-2440, BD Biosciences, Cat No 562681) (1/50), CD11c (Percp.Cy5.5, clone N418, BioLegend, Cat No 117327) (1/20), CD11b (V500 clone M1/70 BD Biosciences, Cat No 532127 or pacific blue, clone M1/70, BD Biosciences, Cat No 532681) (1/50), MERTK (FITC, clone 2B10C42, BioLegend, Cat No 151503) (1/50), CD64 (PE, clone X54-5/7.1, BioLegend, Cat No 139303) (1/20), MHC II (FITC clone AF6-120.1, BD Biosciences, Cat No 562011 or PE-CF594, clone AF6-120.1, BD Biosciences, Cat No 562824) (1/50), CD45.1 (FITC, clone A20, BioLegend, Cat No 110705) (1/50), CD45.2 (APC-Cy7, clone 104, BioLegend, Cat No 109823) (1/50), CD206 (APC, clone C068C2, BioLegend, Cat No 141707) (1/20), CD40 (Pe-Cy5, clone 3/23, BioLegend, Cat No 124617) (1/20). Rat IgG2a, κ Isotype Ctrl Antibody (APC, Clone RTK2758, BioLegend, Cat No 400511) (1/20), Rat IgG2a, κ Isotype Ctrl Antibody (PE-Cy5, Clone RTK2758, BioLegend, Cat No 400509) (1/20). Cells were routinely stained with 7-AAD (BioLegend Cat No 420403) (1/100) or Sytox (Thermo Fisher Scientific, Cat No S34862) (1/1000) for dead cell exclusion. All fluorochrome-labeled antibodies were from BioLegend unless otherwise mentioned. FACS analyses were performed using a BD LSRFortessa flow cytometer (BD Biosciences) and cell sorting was performed on a BD AriaIII. The purities of sorted cells were ≥90% in all sorted samples. Data were analyzed using FACS Diva and FlowJo software packages. The list of antibodies in Excel format including clone, company name, Cat No and the concentration used, is provided as Supplementary Data 1.

### Isolation of primary murine alveolar epithelial cells (AEC)
Lung homogenates were obtained by instillation of dispase (BD Biosciences) through the trachea into HBSS (Gibco) perfused lungs, followed by incubation (in dispase) for 40 min as previously described[61]. After removal of the trachea and proximal bronchial tree, the lungs were homogenized (GentleMACS, MACS Miltenyi Biotech) in DMEM/ 2.5% HEPES with 0.01% DNase (Serva) and filtered through 100 and 40 µm nylon filters. Cell suspensions were incubated with biotinylated rat anti-mouse CD45 (Clone 30-F11, BD, Cat No 553078), CD16/32 (clone 2.4G2, BD, Cat No 553143) and CD31 (clone MEC13.3, BD, Cat No 553371) mAbs for 30 min at 37 °C followed by incubation with biotin-binding magnetic beads and magnetic separation to deplete leukocytes and endothelial cells prior to further culture. AEC suspensions with a purity ≥90% as determined by FACS were seeded at a density of 120–150,000 cells/cm² in 4-µm–pore size transwells (Corning Inc.), in 24-well plates at a density of 250,000 cells/cm², or in chamber slides (Corning Inc.), and cultured in DMEM enriched with HEPES, L-Glutamine, FCS, and pen/strep. The list of antibodies in Excel format including clone, company name, Cat No is provided as Supplementary Data 1.

### AEC apoptosis
The AEC pellet was resuspended in 10 µl of FC block solution, and after 15 min, Annexin V antibody (Alexa Fluor 647, BioLegend 1:20) diluted in 100 µl Annexin V buffer was added (BioLegend, Cat No 640922). The samples were incubated for 15–25 min at 4 °C. All stained cell suspensions were filtered in 5 ml polystyrene tubes and analyzed by a BD LSRFortessa™ flow cytometer to quantify the proportion of apoptotic AEC.

### AEC proliferation
The AEC pellet was resuspended in 150 µl of diluted permeabilization/ fixation buffer (eBioscience™) and incubated for 30 min at RT. 100 µl of permeabilization/wash buffer (eBioscience™) was added after the incubation period and the cell suspension was centrifuged at 1000 g for 5 min at 4 °C. The supernatant was discarded and 50 µl of Ki67 antibody (11F6, PE, BioLegend) diluted 1:50 in permeabilization/wash buffer was added to each sample for 1 h incubation, in the dark, at 4 °C. After staining, the cells were washed with permeabilization/wash buffer and centrifuged at 1000 g for 5 min. The cell pellet was resuspended in 200 µl of sorting buffer and proliferation was quantified.

### Alveolar leakage measurement
Alveolar barrier leakage was assessed by i.v. injection of 100 µl FITC-labeled albumin (Sigma-Aldrich) and quantification of FITC fluorescence ratios in BALF and serum (diluted 1:100) with a fluorescence reader (FLX 800, Bio-Tek instruments) as described previously[8], and given as arbitraty units (AU).

### Isolation of primary human AEC
Primary human AEC were isolated as described previously[30]. Briefly, the lung was cut into small pieces and washed with HAM's F12 + 10% FCS + 1% Pen/Strep/Amphotericin. Tissue was incubated in dispase overnight and thereafter at 37 °C with gentle rotation for 3 h. The tissue was homogenized in MACS dissociater (GentleMACS, MACS Miltenyi Biotech) and the supernatant filtered through 100, 40 and 20 µm nylon filters. Cell suspension was centrifuged and the pellet was resuspended in DNase-containing medium. As a density gradient media, ficoll was overlaid with 15 ml cell suspension and, afterwards, centrifuged and the interphase was taken out and put into DNase-containing medium. Followed by a centrifugation step and thereafter the pellet was resuspended in cell culture medium. Cell viability was assessed by using Trypan Blue staining. Approximately $3.0 \times 10^5$ AEC (purity ≥90% determined by FACS) were seeded in 24-well inserts with 0.4 µm pore size, and kept in HAM's F12 medium (Biochrom, Berlin, Germany) supplemented with 10% FCS, Pen/Strep and Amphotericin for 7–10 days until confluency. AEC with a purity ≥90% determined by FACS (EpCam⁺ CD45/CD31^neg) were used for further analyses.

### AEC in vitro assays
Primary murine and human AEC were infected with PR/8 at MOI 0.5, as described previously[11]. A/PR8 was diluted in PBS^{-/-} containing BSA and was added to the cells for 1 h, until the inoculum was removed and changed to infection medium (DMEM supplemented with BSA, pen/ strep, L-Glutamine and trypsin) for further incubation. For co-culture experiments, AEC were seeded first, allowed to reach confluence, and were infected with A/PR8 or mock infected. 24 h p.i., BMDM were flow-sorted from the BALF of A/PR8-infected mice (BMDM1 were sorted at D7 p.i., BMDM2 were sorted at D21 p.i.) and added directly to the

monolayer for 24 h. Cells were stained with Annexin V for apoptosis and Ki67 for proliferation assays and quantified by FACS. For western blot analysis, murine AEC were cultured in medium DMEM supplemented with pen/strep, L-glutamine and 10% heat-inactivated FCS. Cells were incubated at 37 °C for 5 days and thereafter treated with rPLET1 (40 ng/ml) for 12 h. Cell lysates were prepared in the same manner, as those used for the kinase activity assay described below. For the proliferation assays, murine AEC were cultured in DMEM enriched with pen/strep, L-glutamine and 2% FCS. Cells were incubated at 37 °C for 4 days and thereafter treated with rPLET1 (40 ng/ml) and/or MEK inhibitor U0126 (Promega, #V1121, 10 μM) for 12 h. For qPCR analysis, murine AEC were cultured in DMEM, pen/strep, L-glutamine and 10% FCS, after reaching the confluence they were infected with PR/8 at MOI 0.5, as described previously[11] and treated with rPlet1 (20–40 ng/ml) and/or Src activator (Santa Cruz Biotechnology, Cat No #sc-3052, 10 μM) for 12 h. Cells were stored RLT buffer and stored in −80 °C for further qPCR analyses. For qPCR analyses of Src family-related genes, murine AEC were cultured as described above and treated with rPlet1 for 1 and 2 h. Cells were taken into RLT buffer and stored in −80 °C for qPCR analysis. In the stimulation assay of TR-AM with the AEC supernatant, AEC were cultured as mentioned above. When the cells reached confluence, they were infected with PR/8 at MOI 2.0 or mock infected, as described above and incubated at 37 °C for 12 h/24 h. The supernatant of these cells was used to stimulate the TR-AM (300,000 cells/cm² in a 24-well plate, seeded in RPMI medium supplemented with pen/strep, L-glutamine, 2% FCS and 2.5% HEPES, 6 h prior to the treatment) for 12 h. The supernatant of TR-AM cultures was used for soluble Plet1 quantification by ELISA.

### Quantitative real-time polymerase chain reaction (qRT-PCR)

Cells were centrifuged at 1400 rpm for 10 min at 4 °C and pellets were resuspended in 350 μl of RLT buffer and stored at −80 °C for RNA isolation. RNA was isolated using RNeasy Kit (QIAGEN) and cDNA synthesized. Quantitative PCR (qPCR) was performed with SYBR green I (Invitrogen) in the AB StepOnePlus Detection System (Applied Bioscience) using the reaction setup provided by the manufacturer's instructions. The following murine primers were used: *β-actin* (FP 5′-ACCCTAAGGCCAACCGTGA-3′; RP 5′-CAGAGGCATACAGGGACAGCA-3′); *Gapdh* (FP 5′-TCCCACTCTTCCACCTTCGA-3′; RP 5′- AGTTGGGATAGGGCCTCTCTT-3′); *Hprt* (FP 5′-ACAGGCCAGACTTTGTTGGAT-3′, RP 5′- ACTTGCGCTCATCTTAGGCTT-3′); *Tgfb1* (FP 5′-CCACCTGCAAGAC CATCGAC-3′, RP 5′-CTGGCGAGCCTTAGTTTGGAC-3′); *Fizz-1* (FP 5′-TCCTGCCCTGCTGGGATGAC-3′, RP 5′-GGCAGTGGTCCAGTCAACGA-3′); *Il1b* (FP 5′-TACCTGTGGCCTTGGGCCTCAA-3′, RP 5′-GCTTGGGAT CCACACTCTCCAGCT-3′); *Inos* (FP 5′-TTGGAGGCCTTGTGTCAGCCC T-3′, RP 5′-AAGGCAGCGGGCACATGCAA-3′); *Arg1* (FP 5′-ACCACAGTCT GGCAGTTGGAAGC-3′, RP 5′-AGAGCTGGTTGTCAGGGGAGTGT-3′); *Tnfa* (FP 5′-CGGTCCCCAAAGGGGATGAGAAGT-3′, RP 5′-ACGACGT GGGCTACAGGCTT-3′); *Mrc1* (FP 5′-GGGACGTTTCGGTGGACTGTG G-3′, RP 5′-CCGCCTTTCGTCCTGGCATGT-3′); *Cd40* (FP 5′-GTTTAAA GTCCCGGATGGA-3′, RP 5′-CTCAAGGCTATGCTGTCTGT-3′); *Cldn1* (FP 5′-CGACATTAGTGGCCACAGCA-3′, RP 5′-TGGCCAAATTCATACCTG GCA-3′); *Tjp1* (FP 5′-GCTTCTCTTGCTGGCCCTAA-3′, RP 5′- GGGA GCCTGTAGAGCGTTTT-3′); *Ocln* (FP 5′-TCTTTCCTTAGGCGACAGC G-3′, RP 5′-AGATAAGCGAACCTGCCGAG-3′); *Plet1* (FP 5′-TCCTC ATCGTCGTCAATCGC-3′, RP 5′-TGAGGCTGAGGGTTGTACTTG-3′); *Yes* (FP 5′-TGAGGCTGCTCTGTATGGTC-3′, RP 5′- GCATTCTGTATCCCCG CTCT-3′); *Lyn* (FP 5′- TGGCTAAGGGTAGTTTGCTGG-3′, RP 5′- CGCAG ATCACGGTGGATGTA-3′); *Src* (FP 5′-GCCTCACTACCGTATGTCC-3′, RP 5′-TTTTGATGGCAACCCTCGTG-3′); *Fyn* (FP 5′-AAGCACGGACGGAAG ATGAC-3′, RP 5′- ATGGAGTCAACTGGAGCCAC-3′). *β-actin* and *Gapdh* expression served as normalization control. The data are calculated as $\Delta C_t = (C_t^{reference} - C_t^{target})$ and given as $\Delta C_t$ (larger $\Delta C_t$ values indicate a higher normalized expression of the target gene) or fold induction of

control ($2^{\Delta\Delta Ct}$). The list of primers in Excel format including sequences, Oligo ID and the company name is provided as Supplementary Data 2.

### Transcriptome analysis by whole-genome microarray

For microarray analysis, different subsets of macrophages at different time points p.i., were sorted as described above. The sorted BMDM1 and BMDM2 were resuspended in 300 μl RLT buffer and stored at −80 °C until RNA isolation and further processing. The isolated RNA was amplified and labeled using the LIRAK kit (Agilent) following the kit instructions. Per reaction, 200 ng of total RNA was used. 2 μg labeled cDNA was hybridized on SurePrint G3 Mouse GE 8×60 K Microarrays (Agilent, Design ID 028005) following the Agilent protocol. Washed slides were dried with acetonitrile, treated with Agilent dye-stabilization solution and scanned at a resolution of 2 μm/pixel with an InnoScan 900 instrument. Image analysis was done with Mapix 6.5.0. Further data analysis was done using R 3.6.1 (R Core Team. R, https://www.R-project.org/2015) and the limma package[62] from Bio-Conductor. Mean spot signals were quantile-normalized. Log-signals of replicate spots were averaged.

### Single-cell capture and library preparation

Following the 10xGenomics library preparation protocol, all samples were sequenced on an Illumina NovaSeq6000 sequencer. We used demultiplexing and the subsequent FASTQ file generation Illuminas bcl2fastq (2.19.0.316) to generate the library. Sequencing reads were aligned to the mouse mm10 reference genome (refdata-gex-mm10-2020-A downloaded from 10xGenomics) using STAR solo (2.7.9a) resulting in a UMI count matrix where cells (barcodes) are columns and rows are genes. During the alignment step UMI deduplication was performed by STAR, as well. STAR was run in Velocyto mode in order to allow for downstream velocity analysis. Droplet filtering was done using the EmptyDrops_CR parameter setting.

### Analyses of single-cell RNA-seq data

Resulting UMI count matrices were pre-processed for further analyses using Scanpy (1.7.2) in a Python (3.8.10) environment[63]. We used scrublet with standard parameters in order to identify the doublets. Overall, 77 doublets were found, evenly distributed across the data; therefore, we did not discard these cells (please see https://github.com/agbartkuhn/Pervizaj-Oruqaj_Plet1_sc_analysis for detailed parameter settings). In order to filter out low-quality cells, doublets, cell debris or ambient RNA the following filter criteria were applied to the data. Cells contain a maximum of 40,000 and a minimum of 1500 UMI counts. Less than 15% of all UMI counts are allowed to map to a mitochondrial origin. Finally, at least 2000 different genes must be detectable (at least one read mapped to the gene) for each cell. In order to reduce the gene set, only those genes were kept, which were expressed in at least 20 cells. UMI counts underwent normalization in respect to the library size and such that each cell contains 10,000 reads. The resulting counts were log transformed. For dimensionality reduction and for data integration with Harmony PCAs were calculated. The data integration with Harmony finished after 3 iterations. A k-nearest-neighbor approach was applied to calculate a neighborhood graph, on which Leiden clustering was performed with resolutions of 0.4, 0.6, 1.0 and 1.4. For further analysis, the Leiden clustered cells with a resolution of 1.4 were used. A two-dimensional visualization of the cells was achieved by calculating UMAPs. Next, automatic cell-type annotation was performed by calculating the average gene expression of all clusters individually. Data was compared to the Mouse Cell Atlas (MCA)[64] using the scMCA R package (https://github.com/ggjlab/scMCA). All clusters belonging to the large central aggregate of cell were annotated as alveolar macrophages and subsequently merged into a single cluster. Differentially expressed genes were identified by the Scanpy rank_gene_groups function. Velocity analysis was done

using scVelo[65]. Gene set enrichment analysis was done using clusterProfiler[66] and fgsea packages in R (R Core Team, 2021) using GO and KEGG annotations. For estimation of connectivity between cell clusters, PAGA (partition-based graph abstraction) algorithm was calculated in order to determine BMDM-to-TR-AM transition using the PAGA function as part of the scVelo package using default settings. Standard settings were applied as described in https://scvelo.readthedocs.io/en/stable/VelocityBasics/ (please see https://github.com/agbartkuhn/Pervizaj-Oruqaj_Plet1_sc_analysis, for detailed parameter settings).

## Lung organoid experiments

Bronchoalveolar lung organoids (BALO) were generated from flow-sorted murine lung epithelial stem cells co-cultured with flow-sorted mesenchymal cells as previously described[36]. In lung organoid cultures, $3 \times 10^5$ WT BMDM (1 or 2) or $4 \times 10^3$ BMDM2 from $Plet1^{flx/flx}$ or $CX_3CR1^{iCre}-Plet1^{flx/flx}$) and organoids were cultured in Matrigel® (Corning) ($\alpha$ MEM, 10% FCS, 100 U/ml penicillin, 0.1 mg/ml streptomycin, 2 mM L-glutamine, 1× insulin/transferrin/selenium, 0.0002% heparin)[37]. In selected experiments, anti-Plet1 Ab (20 ng/ml) or isotype control (20 ng/ml) was added to the organoid medium upon cell seeding. BALO numbers and size were analyzed manually, by light microscopy with an EVOS FL imaging system (Thermo Fisher Scientific) and data are provided as means per each well with single data points representing one well.

## Foci forming assay

For virus quantification, MDCK II cells were seeded in 96-well plates and dilutions for FFU (foci forming unit) (equivalent to plaque-forming units. PFU) quantification were prepared in duplicates from $10^{-1}$ to $10^{-8}$ in PBS/0.2% BSA. Cells were washed once with PBS and infected with 50 µl of virus dilutions. Cells were incubated at 37 °C and 5% $CO_2$ for 45 min, inoculum was removed and cells were covered with 100 µl Avicel medium (2×MEM, 1% Penicillin/Streptomycin, 0.1% NaHCO3, 0.2% BSA, 2 µg/ml Trypsin-TPCK, 1.25% Avicel). Cells were incubated for 30 h at 37 °C and 5% $CO_2$, washed twice, fixed and permeabilized in 4% PFA containing 1% Triton-X-100 for 30 min at RT. Cells were washed with washing buffer (PBS and 0.05% Tween-20) and incubated with 50 µl of primary anti-NP (nucleoprotein) antibody solution (3% (w/v) BSA in PBS$^{+/+}$) for 2 h at RT. After further washing steps, cells were incubated with 50 µl of Horse-Radish Peroxidase (HRP) labeled anti-mouse secondary antibody at RT for 1 h, washed again and plates were left to air-dry and scanned, using the Methods 29 Epson Perfection V500 Photo scan (Epson) at 1200 dpi. The virus titer was calculated according to the dilution factor and is expressed as pfu per milliliter (pfu/ml).

## PLET1 enzyme-linked immunosorbent assay

BAL fluid (BALF) from mice or from IAV-ARDS versus control patients was concentrated using Pierce™ Protein Concentrator PES, 3 K MWCO, 2–6 ml (Thermo Scientific™) and analyzed by commercial PLET1 ELISA (mouse PLET1: Antibody research corporation, USA, human PLET1: MyBioSource, USA) according to the manufacturer's instructions. The absorbance was quantified at 450 nm wavelength in the microplate reader (iMark™, Bio-Rad). Human BALF protein concentration was quantified using Quick Start™ Bradford Protein Assay Kit 2 (#5000202, Bio-Rad), absorbance was quantified at 595 nm wavelength in the microplate reader (iMarkTM, Bio-Rad).

## Histology and immunofluorescence

Lungs were clipped at the trachea before opening of the chest cavity, then perfused with 4% paraformaldehyde (PFA), removed, and then fixed for 24 h in 4% PFA. Lungs were embedded in Paraffin (Leica ASP200S), cut into 5-µm-thick sections, and stained with hematoxylin and eosin (Merck). Images of paraffin lung sections with H&E staining

were quantified using Fiji/ImageJ as follows: from each image, three different regions of interest (ROIs) avoiding bronchial and peri-bronchial regions, were marked. For quantification of alveolar area of each ROI, we measured the total area and upon thresholding (using the built-in Huang segmentation algorithm) the area that is occupied by lung tissue. The % of alveolar space is calculated with the following formula: (ROI_Total_Area - ROI_Tissue_Area)*100 / ROI_Total_Area.

For the analysis of fibrosis, lung sections were stained by Mason's trichrome and the bright field images acquired with the EVOS microscope were corrected for their white balance. Subsequently, for these images, we used a custom-made macro (https://doi.org/10.5281/zenodo.10135192) to perform pixel classification using a pre-trained Ilastik[64] model. This pre-trained model (also available online) classifies all pixels of the input image into 6 different classes (background, non-lung tissue, infiltrative area, regenerative area, connective tissue and healthy tissue). The custom-made macro, opens the bright field images, calls the pixel classification module of Ilastik and then pulls back the classified image. Using the six different intensities of the six classes quantifies the % of each class in relation to the total area of the lung tissue. The macro saves the classified images as well as a tab-separated file with all quantifications from all analyzed images.

For immunofluorescence microscopy, isolated AEC cultured in chamber slides (Nunc) were infected with 0.5 MOI PR8 or mock infected and treated with rPLET1 (CUSABIO, 20 ng/mL in medium) or left untreated. After 12 h of incubation at 37 °C, 5% $CO_2$, the slides were fixed in a 1:1 ratio of cold acetone/methanol for 5 min and blocked with 3% BSA in PBS for 30 min at 4 °C prior to staining. The cells were stained with Alexa Fluor® 594 conjugate ZO-1 antibody (Clone ZO1-1A12; Invitrogen) or mouse PE-IgG1, k isotype control (Clone P3.6.2.8.1; eBioscience™) diluted in PBS$^{-/-}$, 0.1% BSA, 0.2% Triton-X-100 for 2 h, followed by washing and mounting with DAPI containing mounting medium (Vectashield, Vector Labs). For organoid staining, cultures were fixed with 4% PFA for 15 min at RT, washed with PBS$^{-/-}$ and stained with LipidTOX™ neutral lipid staining solution (Thermo Scientific) for 6 h at RT to visualize alveoli. Cultures were washed with PBS$^{-/-}$, co-stained with DAPI and mounted for imaging. Confocal images of organoids were acquired either with a Leica SP5 (software version LAS AF 2.7.3) or a Leica SP8 (software version LAS X 3.5.7) confocal microscope. For the quantification of the organoids (area, diameter and alveoli number), we delineated manually each organoid using Fiji and we also marked manually every alveolus on each confocal z-stack acquired for each organoid.

## Transepithelial resistance

Transepithelial resistance (TER) in murine and human AEC was quantified by a Millicell-ERS2 device. Cells were seeded on 0.4 µm pore size transwells and cultured until achieving electrochemical resistances of ≥600 Ω /cm² in murine and ≥400 Ω /cm² in human AEC. Cells were infected with PR8 at MOI 0.5 or mock infected for 1 h at 37 °C and then supplied with recombinant murine (20 ng/ml) or human (40 ng/ml) PLET1 (CUSABIO Technology LLC) in medium or left untreated. The TER was recorded at the indicated time points and is given as Ω/cm².

## Peptide-based kinase activity assay (phosphoproteome)

AEC were cultured until they reached confluence. rPlet1 treatment (20 ng/ml) was started 12 h prior to protein extraction for peptide-based kinase activity assay and western blot analysis. For protein lysate preparation, we placed cell cultures, grown in petri dishes (10 cm diameter dishes), on ice, and washed the cells twice with 5–6 ml cold PBS. Then we added 170 µl M-PER lysis buffer (Thermo Fisher Scientific, Waltham, MA, USA) containing phosphatase and protease inhibitors (Pierce, Rockford, IL, USA). Cells were scraped and homogenized at least 3 times using a syringe-and-needle (20-gauge). The lysate was incubated for 1 h at 4 °C on rotating shaker followed by centrifugation at 16,000 × g for 15 min. The supernatant containing

proteins was aliquoted and immediately flash-frozen in liquid nitrogen and stored at −80 °C. Protein concentration was determined according to the manufacturer's instructions using bicinchoninic acid (BCA) protein assay kit (Thermo Fisher Scientific). Prepared cell lysates were used for peptide-based kinase activity assay and western blot analyses. We performed protein isolation and peptide-based tyrosine kinase activity assay on PTK (phospho-tyrosine kinase) as well as STK (serine/threonine kinase) chips using the PamStation®12 platform (Pamgene, BJ's-Hertogenbosch, The Netherlands) as previously described[39]. Instrument operations such as initial processing of samples and arrays as well as image capture were conducted with Evolve12 software (Pamgene, BJ's-Hertogenbosch, The Netherlands). Numerical values of the individual phospho-peptide spot intensity were obtained by software-assisted densitometric quantification. Background reduction and log2-transformation were done prior normalization (i.e., centering) for the two independent runs. Individual signatures of substrate peptide phosphorylation for each experimental condition (with or without rPlet1 treatment) are compared with databases containing empirical in vitro/in vivo as well as literature-based protein modifications.

Kinases that show a significant change in activity between the two experimental conditions (w/o rPlet1 vs rPlet1 treatment) are described by two important parameters: the "normalized median kinase statistics" (i.e., the predicted differential kinase activity) that depicts the overall change of the peptide set that represents the group of substrates for the given kinase, and the "mean specificity score", which is expressed as the negative $\log_{10}$ p value, where $p < 0.05$ refers to the statistical significance for the changes of the phosphorylation for the substrate peptide sets between the two experimental conditions. Therefore, predicted kinases with a high differential kinase activity (i.e. "normalized median kinase statistics") and a "mean specificity score" higher than 1.3 were considered as promising candidates for subsequent investigations.

## Western blot

Protein lysates were performed as described above. Equal protein amounts (30 μg) were added into each well. Proteins were separated by SDS-PAGE and transferred to nitrocellulose membranes for 60 min at 100 V. Membranes were blocked with 5% non-fat dry milk in PBS/Tween-20 for 1 h at RT. Thereafter, membranes were incubated with the primary antibody for 1 h at RT or at 4 °C o/n, in PBS/Tween-20 with non-fat dry milk, on a shaker. Primary antibodies were obtained from Cell Signaling Technology (Danvers, MA, USA) unless otherwise stated. The following primary antibodies were used: c-Raf (#9422, 1:1,000 dilution), p44/42 MAPK (Erk1/2) (#4695, 1:1,000 dilution), phospho-p44/42 MAPK (Erk1/2) (#4370, 1:1,000 dilution), phospho-c-raf (#9427, 1:1,000 dilution), GAPDH (#2118, 1:2,000 dilution). Membranes were washed with PBS/Tween-20 for 5 min, 3×, on a shaker. The HRP-linked secondary antibody (#7074S, Cell Signaling Technology) was diluted 1:2000 in 2% dry milk with PBS/Tween-20 and incubated with the membrane for 1 h at RT. Membranes were washed with PBS/Tween-20 for 5 min, 3×, on a shaker. Protein bands were visualized using SuperSignal™ West Femto Maximum Sensitivity Substrate (Thermo Fisher Scientific) and detected using the enhanced chemiluminescent (ECL) western blotting system (GE Healthcare, München, Germany). The list of antibodies in Excel format including clone, company name, Cat No and the concentration used, is provided as Supplementary Data 1.

## Cytokine quantification

We quantified concentrations of selected cytokines and chemokines in BALF of mice using a BioPlex MAGPIX Multiplex Reader (BIO-RAD, United States) according to the manufacturer's instructions. Data were analyzed with BioPlex Data Pro software.

## Statistics

Data are shown in scatterplots as single data points. Mean ± SEM per group is indicated by bars and error bars. We used two-sided Student's test to compare between two groups, performing Welch's correction when sample distribution was significantly different between the compared groups. For the analysis of three or more groups, we used ANOVA followed by Tukey's post-test or Brown Forsythe and Welch ANOVA when samples presented different distributions. We used two-way ANOVA to compare samples presenting two variables followed by Sidak's post-test. We used Spearman's rank correlation test to detect a significant correlation in Fig. 7c, and Log-rank (Mantel–Cox) test to analyze survival curves. The statistical test used to address significant differences between groups is described in the legend of every graph and also the exact p values and n numbers. Graphs were prepared using GraphPad Prism (GraphPad Software version 9.3.1).

## Reporting summary

Further information on research design is available in the Nature Portfolio Reporting Summary linked to this article.

## Data availability

The microarray data used in this publication have been deposited in NCBI's Gene Expression Omnibus[65] and are accessible through GEO Series accession number GSE208000. The scRNA-seq data have been deposited in NCBI's Gene Expression Omnibus[65] and are accessible through GEO Series accession number GSE208294. Source Data are provided with this paper.

## Code availability

Code for the scRNA-seq analysis is accessible at https://github.com/agbartkuhn/Pervizaj-Oruqaj_Plet1_sc_analysis. The custom-made macro and the pre-trained model that was used to quantify the fibrosis in lung sections are freely available online (https://doi.org/10.5281/zenodo.10135192).

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

## Acknowledgements

This study was funded by the German Research Foundation (DFG; KFO309 project number 284237345 to S.H., R.E.M., I.V. and W.S.; SFB-TR84 project number 114933180 to S.H.; SFB1021 project number 197785619 to S.H. and J.W.; EXC2026 project number 390649896 to S.H., I.V. and W.S.), the German Ministry for Education and Research (BMBF, grant IPSELON to S.H.), the German Center for Lung Research (DZL; to S.H., R.E.M., I.V., R.T.S., W.S.), the Institute for Lung Health (ILH; A.I.V.A., S.H., W.S.), and the von Behring Röntgen Foundation (grant 66-LV07 to I.V.).

We highly appreciate the contribution of Larissa Hamann, Stefanie Jarmer, Maria Gross, Florian Lueck and Melina Cohen for support with animal experiments, human BAL preparation and histology.

## Author contributions

L.P.O., B.S., and M.F. designed and performed experiments, evaluated and interpreted data and wrote the manuscript. J.W., R.D.G. and M.B. performed sequencing and bioinformatic analyses. M.H. and C.M. supported flow cytometry and infection experiments. A.W. and R.T.S. performed and interpreted phosphoproteome analyses. I.A. conducted image analysis on lung sections and interpreted the resulting data. B.W. and S.G. provided human lung tissue. A.I.V.A. designed, performed and interpreted organoid and scRNA-seq experiments. R.E.M., I.V. and W.S. edited the manuscript. S.H. designed experiments, interpreted data, wrote the manuscript, and financed the study.

## Funding

## Competing interests

The authors declare no competing interests.
