## [Peer Review File · Nature Communications]

Alveolar macrophage-expressed Plet1 is a driver of lung epithelial repair after viral pneumoniaREVIEWER COMMENTS

Reviewer #1 (Remarks to the Author):

In this work, Pervizaj-Oruqaj and colleagues showed that Plet1 expressed by macrophages promoted alveolar epithelial cell proliferation and epithelial barrier resealing after influenza infection. Overall, the experimental approaches are pretty comprehensive, and the results are potentially interesting. However, a couple of experiments were not carefully designed with significant caveats, and some conclusions are overstated. Major issues are listed below and need to be fully addressed in the revision.

1. The title is an over-argument. Other mechanisms have been implicated in lung recovery after injury or infection. The title implicates that macrophage-derived Plet1 is the major factor, which is an overstatement. Please revise.
2. The result section lacks critical experimental details for assessing approach and interpretation of the experiments. The authors should provide more experimental details so that the reviewers and readers can assess how the experiments were designed and interpreted.
3. The experimental design and interpretation of Extended Fig. 1e are flawed. Whole body irradiation damages DNA in TR-AMs, which would lead to impaired proliferation of TR-AMs after influenza infection-induced depletion. Therefore, it is not surprising that BMDMs became dominant in the TR-AM pools even at two weeks after infection in this experiment. However, this scenario would not happen in WT mice without irradiation as TR-AM can proliferate in vivo. Two recent reports (Science Immunology 2022, PMID: 35776802; Immunity 2021, PMID: 33951416) using parabiosis or BM transfer into Rag^{-/-}Il2rg^{-/-}KitW/W^v have demonstrated that early Siglefh1 TR-AM pool reconstitution is primarily mediated by TR-AM proliferation (2 weeks after infection). Only after 3 weeks, monocyte-derived AMs start to contribute significantly to the TR-AM pool. Such conclusion is also supported by the scRNAseq data shown in this manuscript. Therefore, Extended Fig. 1e and the interpretation of the data in the manuscript need to be revised or deleted. The authors should discuss their results with the consideration of the data shown in the two reports mentioned.
4. The use of the term BMDM^{inf} and BMDM^{reg} is misleading. These two BMDM subsets are just at two stages toward TR-AM differentiation, rather than in two distinct lineages as the term would hint. Therefore, please just use BMDM1 and BMDM2 as described in the scRNAseq analysis.
5. Extended Fig 3c-f, it is very surprising that Naïve TR-AM transfer were unable to provide the repair function, but aggregated the repair. However, Naïve TR-AM expressed high levels of Plet1 and did increase the Plet1 level in BAL (Figure 4g)? How to explain the discrepancy of the results?
6. Figure 4b and 4c, it would be better to show the dynamics of Plet1 expression in different populations rather than pooled data across all time points. Could the author analyze published lung scRNA-seq (IAV-infection) and examine the expression of Plet1 in other cell populations? I am curious whether macrophages are the main Plet1 producers in vivo or not?
7. What's the macrophage engraftment efficacy? To the reviewer's knowledge, the engraftment efficacy of macrophage transfer is not very high. Furthermore, transfer of cells/liquid into the lung after infection would exacerbate lung pathology due to liquid/cells transferred in a damaged lung.
8. CX3CR1 can be expressed by other cells such as interstitial macrophages and T cells. Please discuss the limitation of using CX3CR1 iCre.
9. What are the virus titer and inflammation levels in the mice? Besides lung injury repair, virus and inflammation levels also affect host morbidity and mortality after influenza infection. These parameters need to be measured and discussed.

Reviewer #2 (Remarks to the Author):

This is an extremely thorough and well presented piece of work.
Some minor comments for clarity:

Lines 168 - just be a bit more clear about which BMDM you are calling which: change to 'In contrast to CD40high

168 pro-inflammatory BMDM1 (termed BMDMinf), transitional CD206high BMDM2. (Add the numbers)

Line 355 - is it really imprinted? Implies it is a trained response.

Figure 2A - I know it is gene names and then proteins, but is there a way to have CD206 instead of MRC1. Just a bit unclear

Figure 3 - the group labelling could be more clear (the legend is a bit hidden under panel J. Maybe put it above. Also open vs closed symbols would help

Reviewer #3 (Remarks to the Author):

Pervizaj-Oruqaj et al performed in-depth studies of the macrophages in the air spaces of mice recovering from influenza A virus (IAV) pneumonia, leading to the discovery that Plet1 is a novel pro-resolving factor produced by monocyte-derived alveolar macrophages (AMs) that is essential to effective restoration of a healthy lung after IAV infection. The studies were well-controlled and robust, including multiple strong and complementary approaches with non-overlapping sets of limitations. Results were effectively communicated. This reviewer was very enthusiastic and feels this manuscript will be a substantial advance. The authors may wish to consider the following.

1. The question remains whether the cell states defined by the authors as BMDM1/2 or BMDMinf/reg are different cells or the same cells at different times in different environments. While not emphatic, the manuscript implies the authors interpret their data as supporting a model in which an early influx of BMDM1/inf cells becomes eliminated (presumably via cell death pathways), followed by a later wave of BMDM2/reg cells which then transition into becoming AM. The data supporting this are scant, mostly RNA velocity analyses. The possibility that the BMDM2/reg cells are derived from BMDM1/inf cells seems at least as likely; the same cells may exhibit different phenotypes at different times, meaning that these represent different transitional myeloid cell states due to differing stages of pulmonary inflammation rather than different myeloid cell-types. This is a hard question to answer definitively, and the reviewers have already forwarded and supported so much new information in this manuscript that this reviewer does not intend to ask for more. However, it is recommended that the authors more clearly communicate that uncertainty may remain over this particular issue.

2. It is not sufficiently clear when different populations of cells being compared were collected for comparisons throughout the manuscript. This reviewer suspects that cells with different surface markers were collected from mice at two different time-points after infection, so that they could be compared to each other. Is that correct? Was this a uniform design element? There seem to be times when both cell types/states are present in the lungs, at least as defined by surface markers (e.g., days 7 to 14, according to extended Figure 2b). When the 2 cell populations are simultaneously present in the same lung, do they exhibit differences described elsewhere in the manuscript?

3. For Figure 2b, how were cells identified as BMDM1 or BMDM2 prior to the assessment of CD40 or CD206 staining shown in this panel? Extended data Figure 2a, suggested as explanatory, only demonstrates a strategy for identifying BMDM, not for separating BMDM1 and BMDM2 as conveyed in Figure 2b. Perhaps the panel is showing alveolar BMDM that were collected in each of two different times?

4. All the heatmaps show cold/low expression being bright red and hot/high expression being dark blue. This is opposite the more customary use of red and blue to denote things that are hot and cold, respectively. This reviewer would suggest switching to the more usual red=hot and blue=cold, or using a different color spectrum that is not so customarily associated with a particular direction.

Reviewer #4 (Remarks to the Author):

IAV infection injures lung epithelial cells and disrupts barrier function. Proper healing of epithelial cells is important to bring the lung to homeostasis. This manuscript demonstrates that BMDMs that transition into tissue resident AMs will heal the epithelial injury by secreting Plet1 and activating growth factors like signaling in alveolar epithelial cells. Plet1 increases AEC recovery and restores the barrier function of epithelial cells. This manuscript opens a new avenue to attenuate viral injury and rescues mice from fatal IAV-induced disease. State-of-the-art techniques are used in this manuscript. Methodical approaches involving both mouse and human organoids, including transgenic mice, and the treatment with recombinant Plet1 are used to prove their hypothesis.

Major Comments

1. It would be helpful to assess the recovery from disease if physiological changes such as oxygen saturation or pressure-volume curve are presented to show improved lung mechanics or function in plet1-treated mice.
2. Lung fibrosis measurements should be made in various time points as the onset and resolution of fibrosis will also be an indication of the resolution of Influenza-induced injury.

Minor Comments

1. For influenza infection, FFU is used as a measuring units. This may not be clear to readers. I would suggest that authors represent this by TCID50 or PFU.
2. Influenza titer or quantity of IAV-mRNA should be measured in mouse and organoid experiments.

Reviewer #5 (Remarks to the Author):

Pervizaj-Oruqaj L et al present an interesting and comprehensive study on lung epithelia repair after infection.

Despite the extensive data generation, I find their analytical strategy lacking. Among the major issues I noticed are the following:

- 1) One begs to ask why they had not used the most recent version of scanpy? They use version 1.7.2, which was deprecated two years ago.
- 2) The authors set a mitochondrial threshold of 20%. However, many studies and the current single cell practices (https://www.sc-best-practices.org/preprocessing_visualization/quality_control.html) have established that mitochondrial counts may reflect biological activity and these harsh filters may exclude important populations from analysis.
- 3) The authors mention that their QC intended to remove doublets and ambient contamination. However, none of the described steps account for these.
- 4) The authors integrate their data using Harmony. However, there are no plots describing what the main batch driver was. Nor there is any justification on why they have selected a method that performs very poorly on data integration (<https://www.nature.com/articles/s41592-021-01336-8>). It will be vital to add scIB (<https://scib-metrics.readthedocs.io/en/stable/>) scores to understand if their data integration is effective.
- 5) The authors need to specify the parameters used in their analysis using scVelo.

Reviewer #1 (Remarks to the Author)

In this work, Pervizaj-Oruqaj and colleagues showed that Plet1 expressed by macrophages promoted alveolar epithelial cell proliferation and epithelial barrier resealing after influenza infection. Overall, the experimental approaches are pretty comprehensive, and the results are potentially interesting. However, a couple of experiments were not carefully designed with significant caveats, and some conclusions are overstated. Major issues are listed below and need to be fully addressed in the revision.

R: We thank the reviewer for his positive evaluation and thoughtful insights, that we think significantly improved our manuscript.

1. The title is an over-argument. Other mechanisms have been implicated in lung recovery after injury or infection. The title implicates that macrophage-derived Plet1 is the major factor, which is an overstatement. Please revise.

R: We thank the reviewer for this suggestion, and changed the title to: “Alveolar macrophage-expressed plet1 is a driver of lung epithelial repair after viral pneumonia”, to not overstate the role of Plet1 over other repair mechanisms.

2. The result section lacks critical experimental details for assessing approach and interpretation of the experiments. The authors should provide more experimental details so that the reviewers and readers can assess how the experiments were designed and interpreted.

R: We appreciate the reviewer recommendation and provided the following further details in the methods section:

- We added dilutions of antibodies of flow cytometry throughout the subsection “Flow cytometry and cell sorting” (lines 504-528).
- We added the subsections “AEC apoptosis”, “AEC proliferation”, and “Alveolar leakage measurement” (lines 543, 549, 557).
- We detailed our experimental approach in the subsection “Peptide-based kinase activity assay (Phosphoproteome)” (lines 767-787).
- We added the new subsection single-cell capture and library preparation (line 650).
- We detailed how we acquired confocal images and how organoid imaging was performed in lines 750-755 of the subsection “Histology & Immunofluorescence”
- We included further details on BMDM numbers and culture conditions in the subsection “Lung organoid experiments” (lines 692-695).
- We detailed how doublets were detected in the analyses of scRNA-seq data (665-667)
- We specified the parameters used for scvelo in lines 687-688

3. The experimental design and interpretation of Extended Fig. 1e are flawed. Whole body irradiation damages DNA in TR-AMs, which would lead to impaired proliferation of TR-AMs after influenza infection-induced depletion. Therefore, it is not surprising that BMDMs became dominant in the TR-AM pools even at two weeks after infection in this experiment. However, this scenario would not happen in WT mice without irradiation as TR-AM can proliferate in vivo. Two recent reports (Science Immunology 2022, PMID: 35776802; Immunity 2021, PMID: 33951416) using parabiosis or BM transfer into

Rag-/-Il2rg-/-Kit^{W/Wv} have demonstrated that early Siglefh1 TR-AM pool reconstitution is primarily mediated by TR-AM proliferation (2 weeks after infection). Only after 3 weeks, monocyte-derived AMs start to contribute significantly to the TR-AM pool. Such a conclusion is also supported by the scRNAseq data shown in this manuscript. Therefore, Extended Fig. 1e and the interpretation of the data in the manuscript need to be revised or deleted. The authors should discuss their results with the consideration of the data shown in the two reports mentioned.

R: We thank the reviewer for this valuable comment and have revised our interpretation accordingly. We still felt that the data in Ext Fig 1e was supportive to generally demonstrate that BMDM can restore the TR-AM pool during IAV infection, and that the timepoint where this replenishment starts is associated with the appearance of BMDM2 in our model (i.e., when TR-AM proliferation is virtually abolished). We removed all statements that referred to the predominance of BMDM-derived replenishment over TR-AM proliferation:

- In line 140: We changed from “were the main contributors (89.9 % ± 2.99 % of all TR-AM) to TR-AM replenishment in this model with minor contribution of self-renewing CD45.1+ TR-AM” to, “In line with these data, using CD45.1/2 chimeric mice undergoing total body irradiation, BM transplant, and IAV infection after BM reconstitution (Extended Data Fig. 1e), depleted TR-AM of recipient (CD45.1) phenotype were gradually replenished between D14 and D21 by CD45.2+ BM-derived precursors, confirming that BMDM are able to replenish the TR-AM pool in this model (Extended Data Fig. 1 f, g). ...”
- In line 169: We changed from “Together, these data reveal that apart from TR-AM that are depleted during the infection and widely replenished by BMDM, the latter appear as two distinct phenotypes with defined kinetics following severe IAV infection. In contrast to CD40^{high} pro-inflammatory BMDM (termed BMDM^{inf}), transitional CD206^{high} BMDM are present during the resolution phase of the disease course, and reveal a transcriptional profile associated with tissue regeneration, similar to the TR-AM profile (termed “regenerative” BMDM^{reg}) to, “Together, these data suggest that BMDM recruited to the airways undergo a sequential progression through two distinct phenotypes, with defined kinetics, following severe IAV infection. In contrast to CD40^{high} pro-inflammatory BMDM1, transitional CD206^{high} BMDM2 reveal a transcriptional profile associated with tissue regeneration, similar to the TR-AM profile, and are present during the resolution phase of the disease course, contributing to TR-AM pool replenishment.”
- In line 200: We changed from “...experienced (i.e widely BMDM replenished) TR-AM” to “...experienced (i.e partially BMDM replenished) TR-AM”
- In line 203: we deleted BMDM-replenished
- In line 225: We deleted “(mainly) BMDM^{reg}-derived”

- In line 430 We deleted “minor”.
- In line 425 we deleted: “BMDM-derived”.
- We discussed our data in the context of previous reports (Science Immunology 2022, PMID: 35776802; Immunity 2021, PMID: 33951416) mentioned by the reviewer in lines 428-432: “TR-AM self-renewal has been pointed out as a major contributor to TR-AM replenishment in lung viral infection, being progressively outcompeted by BMDM-derived TR-AM following IAV infection^{3,59}. Although we cannot fully exclude a contribution of self-renewing fetal monocyte-derived TR-AM to Plet1-mediated repair, this seems unlikely since *Cx3cr1^{iCre}-Plet1^{fix/fix}* mice had undetectable levels of Plet1 in BALF.”

4. The use of the term BMDMinf and BMDMreg is misleading. These two BMDM subsets are just at two stages toward TR-AM differentiation, rather than in two distinct lineages as the term would hint. Therefore, please just use BMDM1 and BMDM2 as described in the scRNAseq analysis.

R: Given that BMDMinf and BMDMreg are indeed rather two states of cells than of different lineage, we have changed the terminology to BMDM1 and BMDM2 throughout the manuscript.

5. Extended Fig 3c-f, it is very surprising that Naïve TR-AM transfer were unable to provide the repair function, but aggregated the repair. However, Naïve TR-AM expressed high levels of Plet1 and did increase the Plet1 level in BAL (Figure 4g)? How to explain the discrepancy of the results?

R: We appreciate the reviewer comment and agree that this point warrants further explanation. Regarding the absence of a tissue-protective effect when naïve TR-AM were transferred, we show in Fig. 4g that they are less capable to release high levels of soluble Plet1, as compared to BMDM2 or experienced TR-AM, although they express high levels of Plet1 on transcriptional and protein level (Fig. 4c, Extended data Fig. 6k). We hypothesized that in the macrophage transfer experiments (Fig. 4g), where cells are applied into the lungs of *Ccr2^{-/-}* mice at d3 p.i. and analyzed at d7 p.i., the level of Plet1 release from Naïve TR-AM (around 1500pg/ml, similar as the release of low-Plet1-expressing BMDM1) is not sufficient to prevent injury or drive repair of the alveolar epithelium. We speculate, that release/shedding of Plet1 requires further signals from the lung microenvironment that are present later in the disease course (i.e. in the repair phase at and after d21). This would explain why only macrophages isolated at d21 or later from infected WT mice can release high amounts of Plet1 in the recipient lung, and transfer of naïve TR-AM into the alveolar microenvironment early in the disease course (d3 p.i.) does not induce strong Plet1 release due to lack of such signals. We are currently focusing on the molecular mechanisms of Plet1 transcriptional regulation and mechanisms of release, but we think that a deeper analysis would be beyond the scope of this manuscript. However, when we added an amount of 2000 µg of rPlet1 together with naïve TR-AMs to reach Plet1 levels released from experienced TR-AM (Fig. 4g), we could restore the anti-apoptotic and pro-proliferative effects that are observed in experienced TR-AM (6,1 ± 0,9 and 10 ± 1,4; data not shown).

Also, we corroborated that naïve TR-AMs do not aggravate the injury as we showed that levels of apoptosis, alveolar leakage and proliferation are not significantly different between transfer of naïve TR-AMs or a “cell transfer control” constituted of 3T3 cells (see response to this reviewer’s question 7). The data are integrated in new Ext. Fig. 3c-f.

We included a brief comment in lines 423-427: “Interestingly, naïve, non-replenished TR-AM expressed Plet1, but did not release high amounts upon transfer into inflamed/injured lungs of IAV-infected *Ccr2*^{-/-} mice, whereas transfer of BMDM2 or experienced TR-AM resulted in highly increased Plet1 BALF concentrations. This could explain why BMDM2/experienced TR-AM transfer, but not naïve TR-AM transfer, protected the lung epithelium of IAV infected *Ccr2*^{-/-} mice.”

6. Figure 4b and 4c, it would be better to show the dynamics of Plet1 expression in different populations rather than pooled data across all time points. Could the author analyze published lung scRNA-seq (IAV-infection) and examine the expression of Plet1 in other cell populations?

I am curious whether macrophages are the main Plet1 producers in vivo or not?

R: According to the reviewer’s suggestion, we added data from different timepoints in the different macrophage populations (new Fig. 4b, c).

We also addressed Plet1 expression in other lung cell types. We decided to use our own datasets because the experimental settings (influenza virus strain, infection dose, mode of infection, mouse lines etc) are the same and therefore well comparable. We first analyzed Plet1 expression under homeostasis (non-infected, d0) in whole mouse lung CD45-neg cells (unpublished scRNA-Seq dataset from our group) and revealed Plet1 expression predominantly in ciliated airway cells. Furthermore, we analyzed Plet1 expression from whole lung cells FACS-enriched for CD45⁺ cells (unpublished scRNA-Seq dataset from our group) at d0 (non-infected), d3 and d7 after IAV infection and reveal expression in TR-AM but not in other leukocyte subsets of whole lung tissue. These transcriptomic data were complemented by new experimental data using flow cytometry from Plet1-reporter mice (*Plet1*^{fix/fix}) at d0, 7 and 21 p.i. confirming transcriptomic data. TR-AM Plet1-tomato MFI is given as comparison (all data shown below). We added a brief comment on this to the discussion section (lines 432-436). Due to space restrictions in the text file, we did not add these further data to the manuscript.

- Plet1 expression in CD45⁻ cells

a.

b.

a, Dot plot depicting basal *Plet1* expression in CD45⁻ cells of lung tissue. Data obtained from scRNA-seq analysis of whole lung cells. Numbers beneath bars represent whole cell number of the respective population, circle size depicts fraction of *Plet1*-expressing cells within the population, colour indicates mean expression in group. **b**, Flow cytometry data showing *Plet1* expression on Alveolar Type 1 (CD45CD31--EpCamlowpro-SPCnegT1alpha+) and Type 2 cells (CD45-CD31-Epcamlowpro-SPC+T1alpha-); CD45-CD31-EpcamhighCD24low (containing club cells and bronchoalveolar stem cells) and CD45-CD31-EpcamhighCD24high cells (ciliated cells) obtained using *Plet1* reporter mice on days 0, 7 and 21 post IAV infection. Gatings were performed according to Quantius et al, PLoS Pathog, 2016 (DOI: [10.1371/journal.ppat.1005544](https://doi.org/10.1371/journal.ppat.1005544)).

- Plet1 expression in CD45⁺ cells

a. D0

b. D3 p.i.

c. D7 p.i.

Dot plot depicting *Plet1* expression in CD45⁺ cells of lung tissue, on **a**, d0, **b**, d3 p.i. and **c**, d7 p.i.. Data obtained from scRNA-seq analysis of lung CD45⁺ cells. Numbers beneath bars represent whole cell number of the respective population, circle size depicts fraction of Plet11-expressing cells within the population, colour indicates mean expression in group. **D**, Expression of Plet1 in CD45⁺ cells of lung tissue on days 0, 7 and 21 post IAV infection. The results were obtained by flow cytometry analysis on Plet1 reporter mice and expressed as mean fluorescence intensity. Gatings and definition of leukocyte populations were performed according to the gating strategy published by Misharin et al, 2013 (DOI: 10.1165/rcmb.2013-0086MA).

7. What's the macrophage engraftment efficacy? To the reviewer's knowledge, the engraftment efficacy of macrophage transfer is not very high. Furthermore, transfer of cells/liquid into the lung after infection would exacerbate lung pathology due to liquid/cells transferred in a damaged lung.

We agree with the reviewer's comment that macrophage engraftment after intrapulmonary transfer is generally low, but can be increased when macrophage niches have been "emptied" before, e.g. via clodronate-mediated depletion (Guilliams M and Scott CL, 2017. 10.1038/nri.2017.42). In our model, we aimed at placing the macrophages into the lungs of IAV-infected mice to directly make them exert their tissue-protective/reparative function (between d3 and d7 p.i.), not requiring long(er)-term engraftment, but just local release of repair factors, in our case, Plet1. Considering animal ethics aspects and expected gain of knowledge we refrained from pursuing further systematic experimental analyses of engraftment efficiency of the seven different transferred macrophage populations over time, requiring a substantial amount of infection experiments in WT donor and in *Ccr2*^{-/-} recipient mice. Furthermore, we agree with the reviewer that cell/liquid transfer to a damaged lung could exacerbate lung pathology. We therefore added a group of a non-macrophage cell line (3T3 cells) transfer (data are integrated into Ext. Fig. 3d-f) to control for this effect.

8. *CX3CR1* can be expressed by other cells such as interstitial macrophages and T cells. Please discuss the limitation of using *CX3CR1* iCre.

R: We thank the reviewer and agree that *CX3CR1* conditional KO may have limitations as other cells also show expression of this receptor. However, either when we analyzed *Plet1* transcript expression in scRNAseq analysis or when we quantified Plet1 protein expression by flow cytometry in IAV-infected mice (see response to question 6 by this reviewer) we could not detect significant Plet1 expression in T cells, DCs or interstitial macrophages that may all express *CX3CR1*. We therefore think that, even if *CX3CR1* might not be limited to BMDMs

only, this would not substantially affect the BMDM-specificity regarding Plet1 expression in our model.

Nonetheless, we added a brief discussion of the limitations of CX3CR1 approach in line 432: “The possibility exists that Plet1 knockout in other CX3CR1-expressing cells contributes to the outcome of *Cx3cr1^{iCre}-Plet1^{flx/flx}* mice. For instance, interstitial macrophages, DCs and T cells can express CX3CR1; however, we found no significant levels of Plet1 expression in those cell populations in scRNAseq or flow cytometry data (data not shown).”

9. What are the virus titer and inflammation levels in the mice? Besides lung injury repair, virus and inflammation levels also affect host morbidity and mortality after influenza infection. These parameters need to be measured and discussed.

R: We have added new data on virus titers (quantified as pfu/ml) and cytokines from BALF, and histologic quantification of inflammatory infiltrates from both the *Cx3cr1^{iCre}-Plet1^{flx/flx}* loss of function and the rPlet1 treatment models. The data reveal that there is no significant difference in virus titers at d7 p.i. in the Plet1 vs. PBS treatment groups in BALF, and no detectable virus counts in d10, d14 and d21 *Cx3cr1^{iCre}-Plet1^{flx/flx}* vs *Plet1^{flx/flx}* mice (not shown), suggesting no significant effect of Plet1 on virus clearance. As to the inflammatory responses, we reveal that application of rPlet1 early in the disease course (at d3 p.i.) does not affect inflammatory cytokines in BALF at d7, whereas a delay in the resolution of inflammation was found at d10 when Plet1 was deleted in BMDM2. New data are integrated in the revised manuscript in Extended Data Fig 6n-p and Extended Data Fig 7a-d and the related results paragraph.

Reviewer #2 (Remarks to the Author):

This is an extremely thorough and well presented piece of work. Some minor comments for clarity:

1. *Lines 168 - just be a bit more clear about which BMDM you are calling which: change to 'In contrast to CD40high pro-inflammatory BMDM1 (termed BMDMinf), transitional CD206high BMDM2. (Add the numbers)*

R. We thank the reviewer for the encouraging statement on our manuscript and, according to his advice, have changed the text in line 171 of the current version as follows: “In contrast to CD40high pro-inflammatory BMDM1, transitional CD206high BMDM2 reveal a transcriptional profile associated with tissue regeneration, similar to the TR-AM profile, and are present during the resolution phase of the disease course, contributing to TR-AM pool replenishment”. According to the suggestion by reviewer 1 (comment nr. 4) we have also renamed BMDMinf as BMDM1 and BMDMreg as BMDM2 throughout the manuscript.

2. *Line 355 - is it really imprinted? Implies it is a trained response.*

R. The reviewer is right, we do not address trained immunity or epigenetic imprinting in macrophages, and therefore rephrased in line 366 and line 369 in the revised version as follows:

“Our data for the first time provide evidence that a specific epithelial repair program is induced in alveolar macrophages, via a distinct trajectory of functional BMDM specification

from pro-inflammatory/injury-promoting to tissue-healing, with the latter effects mediated by Plet1. The BMDM2 Plet1-driven epithelial-protective phenotype remained upregulated in replenished. ...”

3. *Figure 2A - I know it is gene names and then proteins, but is there a way to have CD206 instead of MRC1. Just a bit unclear*

R: We added “coding for CD206” next to Mrc1 in the figure 2a caption.

4. *Figure 3 - the group labelling could be more clear (the legend is a bit hidden under panel J. Maybe put it above. Also open vs closed symbols would help.*

R: We appreciate the reviewer suggestion and rearranged the group labelling in figure 3 accordingly.

Reviewer #3 (Remarks to the Author)

Pervizaj-Oruqaj et al performed in-depth studies of the macrophages in the air spaces of mice recovering from influenza A virus (IAV) pneumonia, leading to the discovery that Plet1 is a novel pro-resolving factor produced by monocyte-derived alveolar macrophages (AMs) that is essential to effective restoration of a healthy lung after IAV infection. The studies were well-controlled and robust, including multiple strong and complementary approaches with non-overlapping sets of limitations. Results were effectively communicated. This reviewer was very enthusiastic and feels this manuscript will be a substantial advance.

R: We thank the reviewer for his enthusiastic statement on our manuscript and are happy to provide the following answers to his/her questions:

The authors may wish to consider the following.

1. *The question remains whether the cell states defined by the authors as BMDM1/2 or BMDMinf/reg are different cells or the same cells at different times in different environments. While not emphatic, the manuscript implies the authors interpret their data as supporting a model in which an early influx of BMDM1/inf cells becomes eliminated (presumably via cell death pathways), followed by a later wave of BMDM2/reg cells which then transition into becoming AM. The data supporting this are scant, mostly RNA velocity analyses. The possibility that the BMDM2/reg cells are derived from BMDM1/inf cells seems at least as likely; the same cells may exhibit different phenotypes at different times, meaning that these represent different transitional myeloid cell states due to differing stages of pulmonary inflammation rather than different myeloid cell-types. This is a hard question to answer definitively, and the reviewers have already forwarded and supported so much new information in this manuscript that this reviewer does not intend to ask for more. However, it is recommended that the authors more clearly communicate that uncertainty may remain over this particular issue.*

R: We acknowledge and appreciate the reviewer recommendations. Indeed, a definitive answer about BMDM phenotype dynamics represents a major technical challenge, and might be beyond of the scope of the present work. Nevertheless, we agree with the reviewer that the data hardly indicate that two waves of BMDMs are being recruited to the lungs but rather

suggests that at least a fraction of BMDMs transits from BMDM1 CD40high to BMDM2 CD206high. As it was not our intention to support the idea that two different waves of BMDM were recruited to the lungs, we made the following modifications to the text in order to clarify our interpretation of the results regarding BMDM phenotype dynamics.

- We decided to unify BMDM phenotypes maintaining the terms BMDM1 and BMDM2 throughout the manuscript for the sake of clarity and to emphasize the progressive change of phenotype, rather than pointing at two “lineages” (see also response to reviewer #1 question 4).
- In line 169 we modified the text as follows: Together, these data suggest that BMDM recruited to the airways undergo a sequential progression through two distinct phenotypes, with defined kinetics, following severe IAV infection. In contrast to CD40high pro-inflammatory BMDM1, transitional CD206high BMDM2 reveal a transcriptional profile associated with tissue regeneration, similar to the TR-AM profile, and are present during the resolution phase of the disease course, contributing to TR-AM pool replenishment.
- In line 178 we modified the text as follows:” To investigate whether BMDM transition towards BMDM2 endowed the latter with an epithelial-regenerative phenotype, as suggested by the transcriptome analysis, flow-sorted BALF BMDM subsets were co-cultured with ex vivo IAV-infected murine alveolar epithelial cells (AEC)”
- In line 372 of the discussion section, we added the following sentence “Also, despite our RNA velocity data on a likely *in vivo* transition of BMDM1 to BMDM2 phenotype, we cannot exclude that an additional wave of BMDM2 enters the lungs at later stages of the infection course, when repair of the lungs starts to be initiated.”

2. It is not sufficiently clear when different populations of cells being compared were collected for comparisons throughout the manuscript. This reviewer suspects that cells with different surface markers were collected from mice at two different time-points after infection, so that they could be compared to each other. Is that correct? Was this a uniform design element? There seem to be times when both cell types/states are present in the lungs, at least as defined by surface markers (e.g., days 7 to 14, according to extended Figure 2b). When the 2 cell populations are simultaneously present in the same lung, do they exhibit differences described elsewhere in the manuscript

R: Indeed, the reviewer is right, all the experiments comparing BMDM1 and BMDM2 phenotype and *ex vivo/in vivo* functional phenotype were performed with BMDM1 isolated from BALF of IAV infected mice on day 7, and BMDM2 isolated on day 21 post infection, to use the “extremes” of functional polarization. We included additional details in the results section (and figure legends) to add clarity over this point.

- **Line 158:** “sorted CD40high versus CD206high BMDM, obtained on days 7 and 21 p.i. respectively.”

- **Line 181:** “While BMDM1 (collected at d7 p.i.) increased IAV-induced AEC apoptosis, BMDM2 (collected at d21 p.i.) induced AEC proliferation after infection (Fig. 3a, b).”
- **Line 1380:** “qPCR validation of genes expressed in BMDM1 obtained on d7 vs. BMDM2 obtained on d21, or BMDM1 vs. BMDM2 obtained both on d14, according to DNA microarray profiling shown in Fig. 2d”

We furthermore addressed marker gene expression profiles of d14 BMDM1/2, respectively, supporting our hypothesis on a gradual transition from BMDM1 to BMDM2. We acknowledge that a functional analysis of both populations isolated when simultaneously present in the lung would be supportive to validate our transcriptional findings at functional level in ex vivo or in vivo experiments. It was however very challenging to isolate sufficient numbers of BMDM2 at d14 to conduct the *in vivo* BMDM transfer experiments. We nonetheless isolated BMDM1/2 at d14 p.i. for ex vivo functional studies, however, our results regarding functional characterization were not conclusive, likely because the differences between BMDM1 and BMDM2 at these timepoints are not as distinct as when comparing timepoints where BMDM1 and BMDM2 reveal their full spectrum of pro-inflammatory versus regenerative reprogramming, respectively. We added the comparative d14 marker gene expression profiles of BMDM1 and BMDM2 in the new Ext Fig 2e, to emphasize the gradual transition from BMDM1 to BMDM2 at this level.

3. For Figure 2b, how were cells identified as BMDM1 or BMDM2 prior to the assessment of CD40 or CD206 staining shown in this panel? Extended data Figure 2a, suggested as explanatory, only demonstrates a strategy for identifying BMDM, not for separating BMDM1 and BMDM2 as conveyed in Figure 2b. Perhaps the panel is showing alveolar BMDM that were collected in each of two different times?

R: Independently of the timepoint of collection, we first applied the gating strategy depicted in Ext. Fig. 2a, then subgated the BMDM regarding CD40/CD206 expression as depicted in Ext. Fig. 2b. Fig. 2b is indeed showing alveolar BMDM collected in two different time points. To be more precise regarding the histograms depicted in figure 2b, we added this information in the figure legend (**line 1202**): “b, Representative FACS histograms displaying CD40 expression in BMDM1 and CD206 expression in BMDM2 collected from BALF of IAV infected mice on days 7 and 21, respectively. In line **1209** we added: “...prior gating of BMDM was performed according to Extended Data Fig. 2a”.

4. All the heatmaps show cold/low expression being bright red and hot/high expression being dark blue. This is opposite the more customary use of red and blue to denote things that are hot and cold, respectively. This reviewer would suggest switching to the more usual red=hot and blue=cold, or using a different color spectrum that is not so customarily associated with a particular direction.

R. We acknowledge the reviewer suggestion and changed the heat maps accordingly.

Reviewer #4 (Remarks to the Author):

IAV infection injures lung epithelial cells and disrupts barrier function. Proper healing of epithelial cells is important to bring the lung to homeostasis. This manuscript demonstrates

that BMDMs that transition into tissue resident AMs will heal the epithelial injury by secreting Plet1 and activating growth factors like signaling in alveolar epithelial cells. Plet1 increases AEC recovery and restores the barrier function of epithelial cells. This manuscript opens a new avenue to attenuate viral injury and rescues mice from fatal IAV-induced disease. State-of-the-art techniques are used in this manuscript. Methodical approaches involving both mouse and human organoids, including transgenic mice, and the treatment with recombinant Plet1 are used to prove their hypothesis. Major Comments:

1. It would be helpful to assess the recovery from disease if physiological changes such as oxygen saturation or pressure-volume curve are presented to show improved lung mechanics or function in plet1-treated mice.

R: We appreciate the reviewer suggestion and agree that it would be interesting to assess lung function during the experiments after soluble Plet1 treatment. Unfortunately, we currently have no possibility to do such experiments under biosafety level 2 conditions, but we envision to implement further preclinical readout parameters that reflect lung mechanics and gas exchange.

2. Lung fibrosis measurements should be made in various time points as the onset and resolution of fibrosis will also be an indication of the resolution of Influenza-induced injury.

R: We thank the reviewer for this valuable suggestion and have quantified the extent of fibrosis by Masson's Trichrome staining in the experiments performed in *Cx3cr1^{iCre}-Plet1^{fix/fix}* versus *Plet1^{fix/fix}* mice and in Plet1-treated vs control-treated WT mice. The data reveal that there are no substantial differences, respectively, and they are now integrated into the revised manuscript as Extended Data Fig. 6p and Extended Data Fig. 7c.

Minor Comments

1. For influenza infection, FFU is used as a measuring units. This may not be clear to readers. I would suggest that authors represent this by TCID50 or PFU.

R: We appreciate the reviewer suggestion and changed ffu to pfu in lines: 477, 715, 1325, and also in Extended Data Fig. 3b. Additionally, in line 703 of subsection "Foci forming assay" on section Materials and Methods, we added the description: "(equivalent to plaque forming units; PFU)"

2. Influenza titer or quantity of IAV-mRNA should be measured in mouse and organoid experiments.

R: We acknowledge the reviewer suggestion and included data on viral titers from infected *Cx3cr1^{iCre}-Plet1^{fix/fix}* versus *Plet1^{fix/fix}* mice and from Plet1-treated vs control-treated WT mice. We found no difference in Plet1-treated vs control-treated WT mice (Extended Data Fig 7d of the revised manuscript), and no detectable virus titers in the *Cx3cr1^{iCre}-Plet1^{fix/fix}* versus *Plet1^{fix/fix}* mice at d10, 14 and d21, whereas lung organoids had not been infected in the presented experiments.

Reviewer #5 (Remarks to the Author):

Pervizaj-Oruqaj L et al present an interesting and comprehensive study on lung epithelia repair after infection. Despite the extensive data generation, I find their analytical strategy lacking. Among the major issues I noticed are the following:

1. One begs to ask why they had not used the most recent version of scanpy? They use version 1.7.2, which was deprecated two years ago.

R: We acknowledge the reviewer appreciation. We used Scanpy 1.7.2 to analyse our dataset as this was the available version when scRNA-seq analysis were performed. Nevertheless, we re-analysed these data using the current Scanpy version 1.9.3. UMAPs derived from both analyses looked almost identical with only extremely minor deviations. We therefore believe that the data analysed with the 1.7.2 Scanpy version is valid (see Figure below).

UMAP embedding derived from Scanpy versions 1.7.2 and 1.9.3 in A) and B), respectively. In both cases a manual clustering depending on cell types is displayed. Variations between the two versions are minimal.

2. The authors set a mitochondrial threshold of 20%. However, many studies and the current single cell practices

(https://www.sc-best-practices.org/preprocessing_visualization/quality_control.html)

have established that mitochondrial counts may reflect biological activity and these harsh filters may exclude important populations from analysis.

R: We thank the reviewer for this comment. Indeed, we flagged cells with 20 % or more mito-counts, but actually did not remove them explicitly from the anndata object. Those cells fulfilled the additional qc metrics (gene counts, read counts, etc.) and are thus correctly retained in the data set. We additionally checked whether these specific cells would cluster in close proximity to each other, indicative of a population with specific functions; however, this was not the case. Therefore, we do not believe that there is an inherent biological specificity to the high mito cells in our dataset (see figure below).

high_mito_counts

UMAP displaying cells with high mito counts (> 20 %) in red color.

3. The authors mention that their QC intended to remove doublets and ambient contamination. However, none of the described steps account for these.

R: We used scrublet with standard parameters in order to identify the doublets. The process of doublet detection is now described in the materials and methods section, lines 665-667. Overall 77 doublets were found. They do not appear to be concentrated in an individual cluster but are found evenly distributed across the data (see Figure below). Therefore, we did not discard these cells.

UMAP displaying doublets as identified with scrublet in red color.

4. The authors integrate their data using Harmony. However, there are no plots describing what the main batch driver was. Nor there is any justification on why they have selected a method that performs very poorly on data integration (<https://www.nature.com/articles/s41592-021-01336-8>). It will be vital to add scIB

(<https://scib-metrics.readthedocs.io/en/stable/>) scores to understand if their data integration is effective.

R: We would like to thank the reviewer for pointing out the scIB utility. We will incorporate the benchmarking process in our future pipelines. Although we tried to run scIB we were not able to introduce Harmony's or BBKNN's output into the scIB data structure. Nevertheless, we believe that harmony's output is trustworthy and in concordance with biological effects. This is based upon the observation that harmony and BBKNN produce highly similar results (see figure below). In addition, we do not expect to see large batch effects (which might potentially pose problems for an integration with Harmony, as it is pointed out in Luecken et al., 2022 (DOI: <https://doi.org/10.1038/s41592-021-01336-8>)). This is because all samples were prepared in the same way (organism, sample preparation, library preparation, sequencing technology) and only the conditions of the mice were different.

UMAP representation of Leiden clustered (res. 1.0) cells after integration with A) Harmony or B) BBKNN.

5. The authors need to specify the parameters used in their analysis using scVelo.

R: We added the information to the materials and methods section in lines 687-688.

Standard settings were applied, as described in <https://scvelo.readthedocs.io/en/stable/VelocityBasics/>.

REVIEWERS' COMMENTS

Reviewer #1 (Remarks to the Author):

The authors have largely addressed my previous comments/concerns.

Reviewer #6 (Remarks to the Author):

The revised manuscript by Pervizaj-Oruqaj and colleagues has now much improved and I am satisfied with the comments and edits provided. There are some minor issues that I think the authors should address prior to publication to ensure reproducibility. I hope the authors find these useful:

- For reproducibility purposes, the authors should specify the default parameters used in Scrublet package.
- For consistency with the point above, the parameters in scVelo should be included. It is possible that the default parameters used for this particular analysis change in future updates of the package, which would affect reproducibility. Therefore, these should be briefly mentioned in the text, in addition to the link to the package.
- The scripts used for analysis should be made available upon publication using an open source repository such as Zenodo or GitHub, and link provided in the text prior to publication

Finally, I would like to congratulate the authors for such considered comments to the concerns raised by all the reviewers. I recommend this manuscript for publication provided the authors address the points above.

- 1.** For reproducibility purposes, the authors should specify the default parameters used in Scrublet package.
- 2.** For consistency with the point above, the parameters in scVelo should be included. It is possible that the default parameters used for this particular analysis change in future updates of the package, which would affect reproducibility. Therefore, these should be briefly mentioned in the text, in addition to the link to the package.
- 3.** The scripts used for analysis should be made available upon publication using an open source repository such as Zenodo or GitHub, and link provided in the text prior to publication.

1-3. Comprehensive details of all parameters are included within the code, as specified in the Code Availability section. The repository containing the code is accessible on GitHub via the following link: https://github.com/agbartkuhn/Pervizaj-Oruqaj_Plet1_sc_analysis.